



# Informing a hydrological model of the Ogooué with multi-mission remote sensing data

Cecile M. M. Kittel[1], Karina Nielsen[2], Christian Tøttrup[3], and Peter Bauer-Gottwein[1]

[1]Department of Environmental Engineering, Technical University of Denmark, Technical University of Denmark, Kgs. Lyngby, 2800, Denmark
[2]National Space Institute, Technical University of Denmark, Kgs. Lyngby, 2800, Denmark
[3]DHI-GRAS, Hørsholm, 2970, Denmark

*Correspondence to:* Cecile M. M. Kittel (ceki@env.dtu.dk)

**Abstract.** Remote sensing provides a unique opportunity to inform and constrain a hydrological model and to increase its value as a decision-support tool. In this study, we applied a multi-mission approach to force, calibrate and validate a hydrological model of the ungauged Ogooué river basin in Africa with publicly available and free remote sensing observations. We used a rainfall-runoff model based on the Budyko framework coupled with a Muskingum routing approach. We parametrized the model using the SRTM DEM, and forced it using precipitation from two satellite-based rainfall estimates, FEWS-RFE and TRMM 3B42 v.7, and temperature from ECMWF ERA-Interim. We combined three different datasets to calibrate the model using an aggregated objective function with contributions from: (1) historical in-situ discharge observations from the period 1953-1984 at six locations in the basin, (2) radar altimetry measurements of river stages by Envisat and Jason-2 at 12 locations in the basin and (3) GRACE total water storage change. Additionally, we extracted CryoSat-2 observations throughout the basin using a Sentinel-1 SAR imagery water mask and used the observations for validation of the model. The use of new satellite missions, including Sentinel-1 and CryoSat-2, increased the spatial characterization of river stage. Throughout the basin, we achieved good agreement between observed and simulated discharge and river stage, with a RMSD between simulated and observed water amplitudes at virtual stations of 0.74 m for the TRMM forced model and 0.87 m for the FEWS-RFE forced model. The hydrological model also captures overall total water storage change patterns, although the amplitude of storage change is generally underestimated. By combining hydrological modelling with multi-mission remote sensing from ten different satellite missions, we obtain new information on an otherwise unstudied basin. The proposed model is the best current baseline characterization of hydrological conditions in the Ogooué in light of the available observations.

## 1 Introduction

River basin hydrology, ecosystem health and human livelihood are intrinsically linked, emphasizing the need for knowledge about hydrological processes at river basin scale. While hydrological models can increase the understanding of the hydrological regime and its vulnerability to changes (Awange et al., 2014), physical observations of hydrological states are required to force and calibrate hydrological models and are crucial to produce useful simulations. Paradoxically, in-situ gauging networks have thinned out over recent decades (Berry et al., 2012). Satellite remote sensing provides a unique opportunity to acquire





information on important components of the land-surface water balance and bridge this gap (Tang et al., 2009). Remote sensing estimates can supplement and, to some extent, replace in-situ observations, where these are insufficient or impossible to acquire (Xie et al., 2012; Knoche et al., 2014). As more remote sensing-based estimates of hydrological variables become globally and publicly available the need for sound and scientifically founded methods to integrate remote sensing observations with

hydrological models increases (van Griensven et al., 2012).

Several studies have benefitted from using remote sensing based estimates to provide hydrological models with necessary basin-scale information and forcing inputs (e.g. Bauer-Gottwein et al. (2015); Stisen et al. (2008); Awange et al. (2016)). A large number of satellite-based products are publicly available, offering gridded, large-scale information at global scale. Furthermore, hydrological models often contain conceptual parameters, which are either impossible or impractical to measure

directly (Xu et al., 2014). In order to estimate the best-fitting parameter values - and subsequently evaluate model performance – model simulations are compared to observations of hydrological variables. Traditionally, hydrological models use discharge measurements to calibrate and validate hydrological models (Bauer-Gottwein et al., 2015; Knoche et al., 2014). However, in many river basins, in-situ data is limited or insufficient (Berry et al., 2012). Instead, remote sensing observations of hydrological state variables such as river level (Schneider et al., 2017), total water storage or soil moisture (Milzow et al., 2011; Xie et al.,

2012; Abelen and Seitz, 2013) can be used to calibrate and validate the hydrological model performance and improve parameter estimation. Alvarez-Garreton et al. (2014) improved model parametrization by exploring multiple hydrological state variables in a multi-objective calibration.

A commonly used supporting dataset is total water storage change inferred from gravimetric remote sensing. Since 2002, the Gravity Recovery and Climate Experiment (GRACE) mission has recorded and mapped temporal anomalies in the Earth's

gravity field. Changes in terrestrial water storage can be inferred from these anomalies. The dataset has been successfully used to evaluate catchment-scale total water storage as part of hydrological model calibration (Xie et al., 2012; Awange et al., 2014; Eicker et al., 2014; Mulder et al., 2015).

The use of radar altimetry to infer river levels is a relatively new field of research as the utility of the observations over narrow water bodies is limited by the footprint of the altimeter and consequent topographical noise (Schumann and Domeneghetti,

2016). Since the 1990s, technological advances and the improvement of retracking algorithms have enabled the extraction of radar altimetry observations of water heights over inland water bodies (Berry and Benveniste, 2013), with accuracies of between 30 and 70 cm – even for rivers less than several hundred meters wide (Villadsen et al., 2015; Schumann and Domeneghetti, 2016). Radar altimetry has been used in several studies to inform hydrological models both for calibration and in data assimilation schemes (Michailovsky et al., 2013; Getirana and Peters-Lidard, 2013; Getirana, 2010). Repeat ground track missions,

such as Envisat or Jason-2 are typically favored in hydrological studies, as time series can be obtained at fixed locations over the river (virtual stations), similarly to traditional gauging stations. Increasing the spatio-temporal resolution of river level observations by applying a multi-mission approach can improve model calibration (Domeneghetti et al., 2014) and the representation of river hydraulics (Tourian et al., 2016).

Therefore, recent studies have focused on densifying the altimetry dataset by incorporating observations from drifting ground

track missions as well (Schneider et al., 2017). The long repeat period results in a higher spatial resolution, as more points





are sampled along the river. Water masks with sufficiently high resolution are required to extract the observations properly. Schneider et al. (2017) used a water mask based on Landsat Normalized Difference Vegetation Index (NDVI) observations to extract CryoSat-2 observations over the Brahmaputra. However, optical data is not suitable in tropical regions with frequent cloud cover. New, publicly available Synthetic Aperture Radar (SAR) observations from Sentinel-1 enable the extraction of

water masks with high spatial and temporal resolution, facilitating the extraction of CryoSat-2 observations over rivers globally. The biggest obstacle in using remote sensing data products in hydrological modelling is the difficulty to define uncertainties in the data (Tang et al., 2009; van Griensven et al., 2012). The latter is still poorly described at a global scale, and current sensors and extraction algorithms are not precise enough to close the water balance based on remote sensing data (Tang et al., 2009). Furthermore, Gebregiorgis et al. (2012) showed a very high correlation between runoff error and precipitation misses

(85%), highlighting the importance of accurate precipitation estimates. Because of the crucial role of precipitation in driving the land-surface water balance, several precipitation datasets are often compared, if possible to in-situ observations, and evaluated through their performance as model input prior to selecting a specific product (e.g. Awange et al. (2016); Milzow et al. (2011); Cohen Liechti et al. (2012); Stisen and Sandholt (2010)).

However, if only one hydrological variable is considered, calibration of the hydrological model can compensate for data

errors, and in turn conceal deficiencies in the model structure. Knoche et al. (2014) and van Griensven et al. (2012) amongst others, stipulate that while remote sensing input data has opened for new possibilities in terms of catchment-scale modelling, calibration focused on discharge observations tends to compensate for input-data errors by compromising the representation of other hydrological processes. Awange et al. (2016) recommend evaluating the sensitivity of multiple outputs (e.g. groundwater recharge or actual evapotranspiration) to assess the effect of different data sets and uncover interdependency between model

evaluation and data. Furthermore, Knoche et al. (2014) identified a correlation between sensitivity to input data errors and model complexity, showing that lumped conceptual models can provide good results in spite of the reduced complexity (Xu et al., 2014). While several studies have investigated the benefits of using a single type of remote sensing data to supplement in-situ data, few studies have combined several remote sensing data types with available in-situ data to inform hydrological models (Milzow et al., 2011).

The choice of model determines the input requirements as well as the level of parametrization, both of which increase with model complexity. Previous studies have used models with varying complexity ranging from fully-distributed physically based hydrologic and hydrodynamic models (Stisen and Sandholt, 2010; Paiva et al., 2011) to semi-distributed models (Xie et al., 2012; Han et al., 2012), and simpler, lumped conceptual rainfall-runoff models (Knoche et al., 2014; Brocca et al., 2010). Whilst gridded remote sensing data offers the possibility to parametrize and drive fully-distributed models with high

spatio-temporal resolution, the choice of model must reflect the user requirements and capacities as well as the availability and uncertainty of the observations used to define the model (Johnston and Smakhtin, 2014). Here, we select a model structure, which can accommodate the integration of different types of remote sensing observations and is suitable in data scarce regions and for a wide range of user requirements.

In this study, we investigate how multi-mission remote sensing observations can be used to inform a hydrological model

of a large ungauged basin, the Ogooué, Gabon. We show how combining multiple, publicly available datasets can increase





the spatio-temporal characterization of river hydrology and improve model parameter definition. Remote sensing observations of precipitation and temperature are used to force the model, and observations of water height and total water storage from satellite altimetry and gravimetric observations respectively are used to supplement historical in-situ discharge observations in the model calibration and validation.

**2   The Ogooué**

The Ogooué is the fourth largest river in Africa by volume of discharge with a mean annual rate of 4700 m$^3$ s. It is 1200 km long and drains approximately 224 000 km$^2$, 90% of which lie within Gabon (Figure 1). The river originates in the Ntalé mountains on the Batéké Plateau in Congo and runs northwest into Gabon. The basin is characterized by plateaus and hills bordering a narrow coastal plain. Although the hills are not very high (mean elevation in the catchment is 450 m), steep slopes

and cliffs several hundred meters above the plain below create characteristic chutes and rapids, and between Lastourville and Ndjolé, the river is unnavigable. After Ndjolé, the river runs west and reaches the 100 km wide and 100 km long Ogooué Delta. The lower part of the Ogooué is navigable and gentler than the rest of the river, with relatively low bed slopes, between 0.07 - 0.13 m km. The river has numerous tributaries. The largest are the Ivindo, which flows from Northeast to Southwest Gabon before draining into the Ogooué just below the Chutes and Rapids of the Ivindo, and the Ngounié, which flows from the Chaillu

Mountains along Gabon's Southern border before joining the Ogooué just upstream of Lambaréné.

The climate is equatorial with two rain seasons: February to May and October to December. Mean annual precipitation is 1831 mm and temperatures vary between 21 and 28°C. The dense vegetation cover across the basin attenuates the potential flooding from the heavy rain in the two rainy seasons, and the basin is not particularly prone to flooding. Large portions of the river are fed by baseflow during the drier austral winter months, when cooler temperatures greatly reduce evapotranspiration

(Mengue Medou et al., 2008).

The main challenge for water resources management in the region is the reconciliation of conservation and development plans. The Ogooué is home to several important ecosystems including several Ramsar sites (i.e. wetlands of international importance) such as the Chutes and Rapids of the Ivindo, Mbougou Baduma and the Doumé Rapids. Conservation of these wetlands is intrinsically linked to the hydrological regime in the basin. The Ogooué also plays a significant role in development

plans in Gabon, both as part of the energy infrastructure and as a transport waterway (World Bank, 2012). Thousands of endemic species have been identified in the region surrounding the Grand Poubara hydropower station and in potential mining sites, and the risk of pollution from mineral industries and transport combined with changes to the flow regime are not negligible for the riparian ecosystems (Mezui and Boumono Moukoumi, 2013).

Hydrological monitoring efforts in the 1960s and 1970s have produced decade-long time series of discharge measurements

at several locations in the basin; however, the most recent publicly available observations are from 1984. Published studies focusing on the hydrological regime of the Ogooué have focused on large scale investigations of African rivers and historical monitoring data (Mahe and Olivry, 1999; Mahe et al., 2013). To the authors' knowledge, there are no previous hydrological modelling studies of the basin.





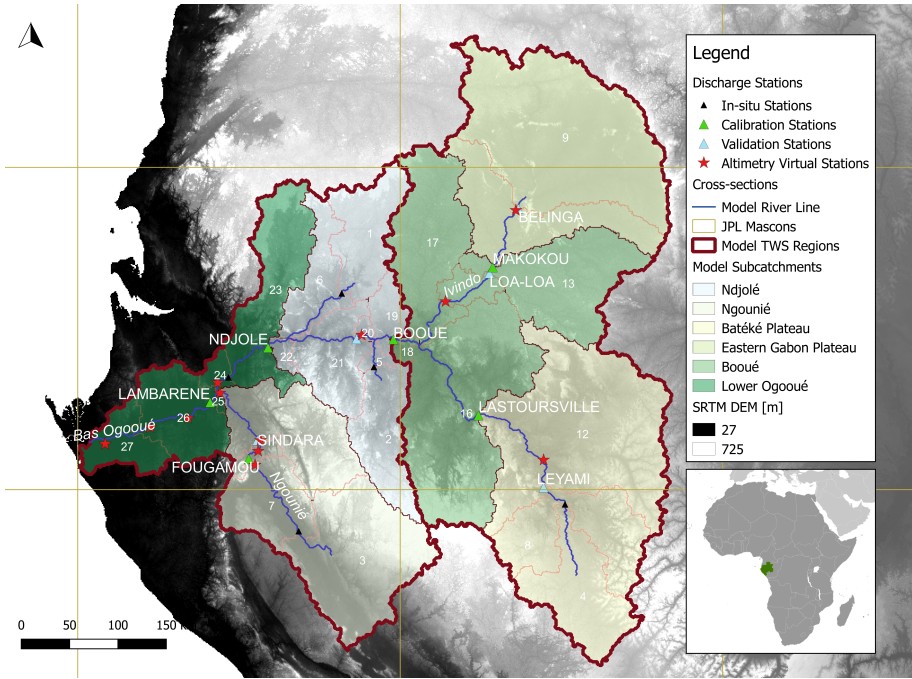

**Figure 1.** Basemap of the hydrological model of the Ogooué basin along with in-situ discharge stations and altimetry virtual stations.

## 3 Data and Methods

### 3.1 Climate Forcing

Daily temperature and precipitation observations are required to force the hydrological model. We used the ERA-Interim re-analysis from the European Centre for Medium-Range Weather Forecasts (ECMWF) as temperature input. Global, 6 hourly

2 m temperature estimates at 0.75 degree spatial resolution can be accessed from 1979 to present with 2 months delay. We select two widely used and well documented satellite-based rainfall estimates to force the model based on results from previous studies comparing satellite rainfall estimates (SRFE) products over the African continent (Thiemig et al., 2013; Stisen and Sandholt, 2010; Awange et al., 2016). The Famine Early Warning System – Rainfall Estimate (FEWS-RFE) has been operational since 2001 and is specifically designed for the African continent. The Tropical Rainfall Measuring Mission (TRMM)

is a global mission launched by NASA in late 1997 and operational until 2015. The TRMM 3B42 v.7 product is a reanalysis product produced from observations from the Global Precipitation Measurement (GPM) since 2015. The dataset has a temporal resolution of 3 hours and a spatial resolution of 0.25 degrees and is provided between 50 degrees south and 50 degrees north. All climate data is aggregated to daily observations. We placed virtual climate stations at the centroid coordinates of each model subbasin and transformed the gridded precipitation and temperature data to point data using zonal statistics over

the subbasins of the hydrological model.





## 3.2 Intercomparison of precipitation data

We compared the two precipitation products in order to identify any significant differences in precipitation trends. The spatial and temporal distribution of rainfall is relatively similar; however, TRMM predicts significantly more rain than FEWS-RFE (1600-2400 mm per year versus 1200-2200 mm). The annual average precipitation and double mass plot (Figure 2, c and d), reveal that while the overall inter-annual variations are similar, the magnitude varies strongly: ranging from nearly similar annual magnitude in 2010 and 2011 to 500 mm more rain in 2006 and 100 mm less rain in 2014 predicted by TRMM compared to FEWS-RFE. A comparison to historical precipitation observations at four locations in the basin revealed that while both products record more days with rain, TRMM is closest to the observed mean monthly precipitation with a RMSD of between 11 and 19% of the observed precipitation compared to RMSD values of 18 to 33% for FEWS-RFE. The satellite-based estimates are gridded data, observing rain events over larger areas than the gauge-stations, thus increasing the probability of recording at least one smaller event every day. Secondly, the period of record differs between the in-situ data and the SRFE observations by over two decades, leaving room for changes in the long-term trends. The analysis indicates the products are relatively similar and we found no large discrepancies in terms of trends between the in-situ and remotely sensed observations. Without up-to-date in-situ precipitation, records covering the entire basin it is impossible to conclude which product best reflects the present precipitation patterns over Gabon. Therefore, we estimate the model parameters using both products as model forcing.

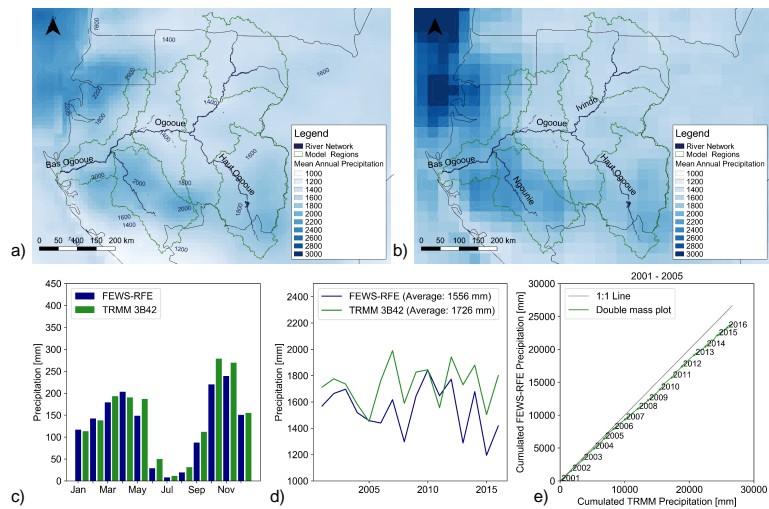

**Figure 2.** Average annual precipitation in the Ogooué basin based on FEWS-RFE (a) and TRMM 3B42 v7 (b), Long-term monthly average (c) and annual average (d) precipitation from TRMM 3B42 v7 and FEWS-RFE v2 and double mass plot (e).

## 3.3 GRACE Total Water Storage

We obtain total water storage observations over the Ogooué from the JPL mascon surface mass change solution applied to Gravity Recovery and Climate Experiment (GRACE) gravimetric observations (Longuevergne et al., 2010; Watkins et al.,





2015). Data from April 2002 to present can be derived at monthly intervals. A mascon-set of multiplicative gain factors are provided with the dataset and can be applied to compensate for the attenuation of small scale mass variations due to the sampling and processing of the GRACE observations – for instance in hydrological studies were these may be significant – by reducing the difference between the smoothed and unfiltered total water storage variations (Long et al., 2015). The gain factors

have a spatial resolution of 0.5 degrees; however, at this resolution the correlation between neighboring cells is much higher. We aggregate the scaled solution to the native resolution of the mascons to produce time series for the two regions within the Ogooué using zonal statistics, splitting the basin along the frontier of two mascons (Figure 1).

### 3.4 SAR Imagery

Sentinel-1 is a two-satellite constellation launched by the European Space Agency (ESA) in 2014 for land and sea monitoring.

The two satellites orbit 180° apart, at a 700 km altitude, ensuring optimal coverage and a short revisit time of 6 days on average. Both Sentinel-1 satellites carry a SAR instrument working in C-band, which penetrates cloud cover. Over land, the satellite operates in Interferometric Wide swath (IW) mode by default, with a swath width of 250 km and a $5 \times 20$ m ground resolution. Sentinel-1 satellites carry dual-polarization SAR instruments, which can transmit and receive signals in vertical (V) and horizontal (H) polarization. In IW mode, dual polarizations VV and VH are available over land.

Level-1 Ground Range Detected (GRD) IW Sentinel-1 images acquired in May and June 2016 over the study area are pre-processed in the ESA Sentinel Application Platform processing toolbox (SNAP). The images are (1) calibrated, (2) speckle filtered using the Refined Lee filter and (3) geocoded using Range-Doppler terrain correction with the 3 s Shuttle Radar Topography Mission (SRTM) DEM as topographic reference. Due to the lower reflectance of water compared to land; the histogram of the filtered backscatter coefficient is expected to contain two peaks of different magnitudes: very low values

of backscatter corresponding to water pixels, and higher values representing the land pixels. The threshold separating water from non-water points is the minimum between the two peaks. We define a threshold value for each individual scene and adjust it manually to ensure the best balance between false positives (where soil moisture enhances absorption thus decreasing backscattering) and false negatives (waves on the water surface enhance reflection and increase backscattering).

### 3.5 Altimetry

We obtain remotely sensed river stages from Envisat and Jason-2 from the River&Lake and Hydroweb project databases (Berry et al., 2005; Santos da Silva et al., 2010) at 12 locations in the basin within the Sentinel-1 water mask, at temporal resolution corresponding to the satellites' return periods: 35 days and 10 days respectively, for the periods 2002-2009 and 2008-2012. Additionally we obtained CryoSat-2 Level 2 data from the National Space Institute, Technical University of Denmark (DTU Space) for the period 17/07/2010 to 21/02/2015. The data provided by DTU Space is based on the 20 Hz L1b dataset provided

by ESA and has been retracked using an empirical retracker. Details concerning data processing are described in (Villadsen et al., 2015). Finally, ICESat laser altimetry observations were obtained from the Inland Water Surface spot heights (IWSH) database for the period 2003-2009. Details on the processing of the ICESat observations can be found in O'Loughlin et al. (2016). The 48 ICESat observations within the Ogooué basin provided on the IWSH database have been filtered using a using





a global water mask and transect-averaged (O'Loughlin et al., 2016). We project all altimetry observations onto the EGM2008 geoid.

We filter the CryoSat-2 observations over the Sentinel-1 river mask using a point-in-polygon approach, and reprojected onto the model river line. 762 CryoSat-2 ground tracks cross the Ogooué basin during the period of record, resulting in 1521

single observations within the river mask. Obvious outliers in the CryoSat-2 dataset are removed using the SRTM DEM. Over the Ogooué, the CryoSat-2 altimeter operates in Low Resolution Mode (LRM). The CryoSat-2 waveform may include topographical noise due to its large footprint in LRM, particularly in the middle part of the river. CryoSat-2 heights are almost consistently smaller than SRTM derived heights. The difference can be attributed to topographical noise and the density of vegetation in the basin, as SRTM contains averaged topography within 90 m pixels and may be recording the top of the canopy.

Furthermore, we identified discrepancies in the longitudinal cross-section of the SRTM DEM along the river line. We reduced the risk of removing potentially valid CryoSat-2 observations based on erroneous SRTM heights by correcting the SRTM heights to the immediate downstream value, if they exceed the upstream elevation by more than 1 m. We define CryoSat-2 outliers as observations more than 20 m lower than the SRTM height or more than 3 m higher. Most outliers are from single observation transects.

River stages from Envisat, Jason-2 and ICESat are transect averaged. In cases where CryoSat-2 overpasses are parallel to the river line, important spatial variations may be lost in a single transect average. However, as most of the Ogooué runs perpendicular to CryoSat-2 satellite tracks, we transect-average the observations to obtain a time series. For tracks crossing subbasin borders, two separate means are calculated. We obtain 524 transect-averaged observations from the 1342 outlier-filtered single observations. Most observations are concentrated in the lower Ogooué (downstream of Ndjolé), furthest downstream of the

river network. Figure 3 shows the longitudinal profile of the SRTM elevation with the ICESat and CryoSat-2 single observations for the entire river (CryoSat-2 outliers are shown in grey). The river network includes confluent branches, resulting in three possible "routes" in the basin: the Ogooué (from the Batéké plateau to the Delta), the Ivindo (from the Eastern Gabon plateau through the confluence to the Ogooué to the Delta) and the Ngounié (from the upstream Ngounié to the Ogooué delta). To ensure each point is associated to a single chainage, we define the latter as the distance of each point on the river to the main

outlet in kilometers. Outliers are concentrated between around chainage 150 and downstream of the Batéké Plateau (chainage 780 on the Ogooué, lower branch in Figure 3). In the upstream regions, the river drops off plateaus and runs through narrow valleys surrounded by steep slopes, increasing the risk of reflections from the surrounding land surface. Visual inspection of the longitudinal profile does not suggest clear bias (see inset in Figure 3); however the ICESat observations are generally larger than the CryoSat-2 observations, which is explained by time of observation: 62.5% of the ICESat observations are sampled

during the wet seasons (February-April and September-December) against only 39.0% of the CryoSat-2 observations.

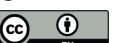


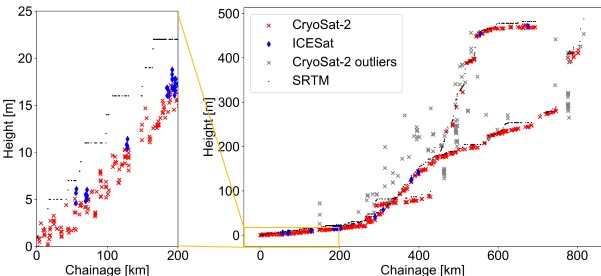

**Figure 3.** Longitudinal profile of the single CryoSat-2 observations, transect-averaged ICESat observations and corrected SRTM reference heights on the Ogooué and its tributaries.

While the altimetry water heights are referenced to EGM2008, the model simulates water depth. To circumvent this discrepancy, we compare water height anomalies. For repeat ground track missions, the average water height recorded at each virtual station is subtracted from each observations to obtain the water height anomalies. Due to the coarse resolution of CryoSat-2 observations, the observations are interpolated over space and time, considering only the day of year (DOY) of the observation. The mean water height at a given chainage is subtracted from the interpolated water heights and from the individual observations to obtain relative water heights or anomalies. The amplitude of Envisat and Jason-2 observations are compared to the CryoSat-2 amplitudes in order to evaluate potential inter-satellite bias throughout the basin (Table 1). Concurrence between the missions strengthens the model evaluation and justifies the multi-mission approach. The interpolated mean annual water elevation at a given chainage is subtracted from the observations and only the day of year (DOY) of the observation is considered. Figure 4 shows the spatio-temporal distribution of the Envisat/Jason-2 observations against the CryoSat-2 observations. The two rain seasons are clearly visible with all missions, with the annual minimum in June-September (DOY 153-244).

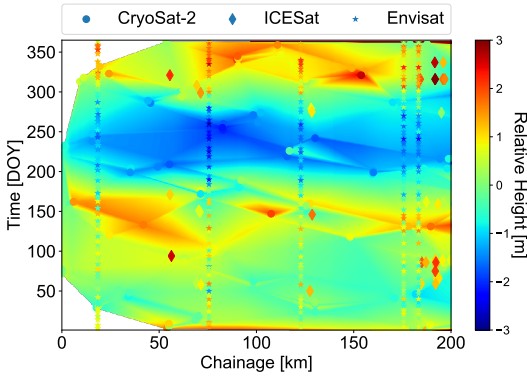

**Figure 4.** Spatio-temporal characterization of the annual water elevation changes of the lower Ogooué.



**Table 1.** Comparison of CryoSat-2 and Envisat/Jason-2 water height amplitudes for three branches of the Ogooué with sufficiently dense CryoSat-2 observations. The dispersion of the amplitudes predicted by CryoSat-2 are given by the standard deviation for the given river section.

|  | CryoSat-2 Observations | CryoSat-2 Amplitude [m] | Virtual Stations | Envisat/Jason-2 Amplitude [m] |
|---|---|---|---|---|
| Upstream of Makokou (Ivindo) | 32 | $3.3 \pm 1.5$ | 1 | 2.22 |
| Upstream of Sindara (Ngounié) | 41 | $2.8 \pm 0.9$ | 3 | 2.4 - 3.2 |
| Downstream of Ndjolé (Ogooué) | 156 | $3.4 \pm 0.7$ | 4 | 2.4 - 3.7 |

## 3.6 Hydrological Model

The hydrologic-hydrodynamic modelling framework used in this study consists of a lumped conceptual rainfall-runoff model based on the Budyko framework and developed by Zhang et al. (2008), coupled to a cascade of linear reservoirs and a Muskingum routing compartment (Chow et al., 1988). Figure 5 shows the model flow chart.

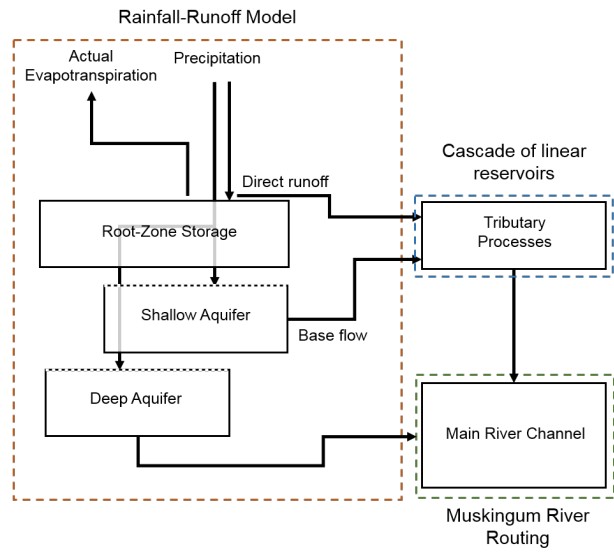

**Figure 5.** Flow chart of the hydrologic-hydrodynamic model along with the two main modifications to Zhang et al. (2008)'s rainfall-runoff model: the deep aquifer and the tributary processes.

5    Zhang et al. (2008) simulate catchment water balance down to a daily time scale using a holistic approach based on the Budyko-framework, which assumes that two parameters control the equilibrium water balance: water availability and atmospheric demand. The former is approximated by precipitation, while the latter is represented through potential evapotranspiration. In Zhang et al. (2008)'s approach, catchment storage is conceptualized as two compartments: root zone storage and groundwater storage. In this study, we add a deep aquifer, splitting groundwater recharge using a simple, time constant par-



titioning coefficient. The two aquifers have different storage constants used to calculate baseflow in the model. At each time step, Budyko's limits concept is used to partition precipitation into direct runoff and catchment rainfall retention, to compute groundwater recharge from the catchment retention and soil storage and to partition soil water availability into actual evapotranspiration (ET) and the updated soil storage. In natural systems, several processes delay direct runoff before it reaches the

main channel (overland flow, transmission losses, evaporation losses, bank storage etc. (Neitsch et al., 2009)) and the basin contains a number of lakes and wetlands, which are not directly resolved by the model. Therefore, we implement conceptual tributary reaches in the form of a Nash cascade of linear reservoirs to route the direct runoff and baseflow from the shallow aquifer to the main channel.

We use Muskingum routing to route discharge from one subbasin outlet node to the next (Chow et al., 1988). The approach

has two parameters: a proportionality coefficient, K, between the cross-sectional area of the flood flow and the discharge at a given section and a dimensionless weighting factor, X. Traditionally, K and X are calibrated using inflow and outflow observations, however in poorly gauged catchments, the parameters can be fitted through calibration and assumptions about channel properties. We estimate K based on segment lengths and average river flow velocity calculated from Manning's equation using trapezoidal cross-sections and a calibrated roughness coefficient (Todini, 2007). In this study, we selected a 1:2 run to rise ratio,

resulting in relatively limited changes in widths. X and Manning's roughness coefficient, n, are calibrated.

### 3.7  Watershed Delineation

We use the SRTM Digital Elevation Model (DEM) and TauDEM watershed delineation hydroprocessing routine (Tarboton, 2015) to derive the drainage network and subbasins. The DEM resolution is reduced to approximately 1 km in order to comply with memory and CPU constraints. We place model outlets at points of interest including in-situ gauging stations and upstream

of key wetlands. The latter are included for reference in future scenario development studies in the catchment. Reach geometry including bed slope, reach lengths and widths are estimated by the hydroprocessing tool and refined based on the Sentinel-1 water mask and a high resolution SRTM DEM. We further subdivide the main channel into reach segments in order to ensure numerical stability of the routing model. We place cross-sections every 5-25 km.

### 3.8  Calibration

In order to include multiple observations of varying spatio-temporal scale, a holistic calibration approach is used. A warm-up period of 1 year allows the model to stabilize. Based on the basin geography, we divide the basin into six calibration zones with common parameter values (Figure 1):

- The Batéké Plateau: The Haut-Ogooué until Lastoursville station (Subbasins 4, 8 and 12)

- The Eastern Gabon Plateau: the upstream Ivindo basin until the Makokou station (Subbasins 9 and 10)

- The Ogooué and the Ivindo catchments until the Booué station (Subbasins 13, 14, 16, 17 and 18)

- The Ogooué until the Ndjolé station (Subbasins 1, 2, 5, 6, 19, 20, 21 and 22)



– The Ngounié (Subbasins 3, 7, 11 and 15)

– The lower Ogooué and Delta until Port Gentil, using the Lambaréné station at the outflow of subbasin 25 for calibration (Subbasins 11, 15, 23, 24, 25, 26, 27)

The calibration parameters are shown in Table 2. In total 60 parameters are calibrated.

**Table 2.** Calibrated model parameters - one set of 10 parameters is defined for each calibration region. Calibration ranges are based on early trials, manual calibration and parameter definitions.

| Parameter Symbol | Description [Unit] | Calibration Range |
|---|---|---|
| $\alpha_1$ | Budyko parameter governing the partition between catchment retention and runoff [-] | $[0.1 - 0.7]$ |
| $\alpha_2$ | Budyko parameter governing the partition between catchment retention and runoff [-] | $[0.1 - 0.7]$ |
| d | Baseflow recession coefficient [day$^{-1}$] | $[0.003 - 0.7]$ |
| $S_{max}$ | Maximum soil water storage [mm] | $[100 - 1500]$ |
| $n_{Nash}$ | Number of identical reservoirs in series in Nash cascade [-] | $[1 - 10]$ |
| $k_{Nash}$ | Reservoir storage constant in Nash cascade [day] | $[1 - 10]$ |
| $X_{GW}$ | Partitioning coefficient of recharge to shallow and deep aquifer [%] | $[0 - 95]$ |
| $d_{deep}$ | Deep aquifer baseflow recession constant [-] | $[0.001 - 0.2]$ |
| X | Muskingum weighting factor [-] | $[0 - 0.5]$ |
| n | Manning's n [-] | $[0.015 - 0.05]$ |

The following sections describe the individual objective functions combined for the calibration as well as the validation of the model.

The hydrological model is calibrated using a global search algorithm, the Shuffled Complex Evolution – University of Arizona (SCEUA) algorithm developed by Duan et al. (1992) and implemented in Python by Houska et al. (2015) in the SPOTPY plugin. The algorithm has been widely used in hydrological studies. The parameters are calibrated by evolving 10

complexes and with convergence criteria of 0.1% change in objective function and parameter value over 100 model runs. We use an aggregated objective function in order to exploit all available and suitable observations in the basin. The objective function contributions minimize the difference between the observed and simulated

– Flow Regimes at Lastoursville, Makokou, Booué, Ndjolé, Fougamou and Lambaréné, using historical observations from the 1930s to the 1980s – the flow regime is characterized by the

– Flow Duration Curves





- The daily or monthly climatology benchmark depending on available observations

- Stages at 12 virtual stations throughout the basin

- Catchment total water storage – due to the coarse resolution of GRACE, the calibration regions are aggregated into two calibration zones upstream and downstream of Booué.

When several objective functions are optimized at once, the optimal solutions representing the trade-offs between the different objectives lie on the so-called "Pareto front". However, it is computationally expensive to compute the full Pareto front for a meaningful number of parameter sets and for high-dimensional problems (Madsen, 2000). Instead, priorities can be given to the individual solutions prior to calibration based on the applications of the model to achieve a compromise between the individual contributions. The aggregated objective function $\phi$, and calibration objective, was defined as the Weighted Root Mean Square

deviation (WRMSD) between the objective function value resulting from the simulation and the objective function value $\phi_{ref,i}$ for a perfect fit.

$$\phi = \sqrt{\frac{1}{N} \sum_{i=1}^{N} (\phi_{ref,i} - \phi_{sim,i})^2 \times w_i} \tag{1}$$

Here, $w_i$ is the weight assigned to each individual objective function contributions. We weigh the observations within the objective functions prior to aggregation in order to account for input-data error and uncertainty. Because all the objective

functions are functions of scaled or weighted residuals, weights of 1 are deemed reasonable for most contributions, except the contributions from GRACE, which are given a weight of 2 to balance the low number of available GRACE observations.

    The goodness-of-fit measures used for each partial objective function are different for the different contributions. We calibrate the FDC based on the method described in (Westerberg et al., 2011). Selected percentiles are chosen based on a discharge volume interval approach. The area under the FDC is divided into 20 equal discharge volume bins with 5% volume increments,

resulting in 19 equally spaced evaluation points. The performance measure is based on a scaled score approach. At a given evaluation point, i, a perfect fit gives a score, S, of 0, while values differing by more than 10% are given scores of 1 and -1 respectively. The performance measure is defined as

$$R_{FDC} = 1 - \frac{\sum_{i=1}^{N-1} |S_i|}{N-1} \tag{2}$$

    The remaining contributions are evaluated based on the WRMSD, using data uncertainty or variability as weights, yielding

the performance measure

$$WRMSD = \sqrt{\frac{1}{N} \sum_{i=1}^{N} \left( \frac{y_{sim,t} - y_{obs,t}}{\sigma_t^2} \right)^2} \tag{3}$$





$\sigma_t^2$ is the standard deviation of the observations for the climatology of day t, and the observation uncertainty for the TWSC and water height contributions. For the water stage comparison, we select a measurement uncertainty of 0.5 m based on previous studies (e.g. Santos da Silva et al. (2010); Birkinshaw et al. (2010)). Villadsen et al. (2015) provide a summary obtained of RMSDs in literature. GRACE measurement uncertainties are provided with the dataset (Longuevergne et al., 2010; Watkins
et al., 2015).

No bathymetry observations are available for the Ogooué; therefore, we compare altimetry water heights to simulated relative water depths. Water depth in the middle of a given reach can be estimated directly from the reach storage, and combined with the water depth of the prism storage to linearly interpolate the water depth along the river line at any distance from the cross-sections.

## 3.9  Sensitivity Analysis

A global sensitivity analysis is carried out based on a Latin-Hypercube Sampling (LHS) of the parameter space. We used the FAST Extended algorithm (Saltelli et al., 1999) implemented in SPOTPY by Houska et al. (2015). FAST provides two sensitivity measures: the first sensitivity index and a total sensitivity index, including contributions from parameter interaction. Over 200 000 model iterations are performed. We use a multi-objective approach, in order to evaluate the sensitivity of the in-
dividual contribution groups to different parameter and identify how including different observation groups constrains different parameters.

## 4  Results and Discussion

### 4.1  Sensitivity Analysis and Parameter Calibration

The sensitivity analysis provides useful information on how the different contributions to the global objective function constrain
different parameters. The sensitivity indices are shown in Figure 6. We find that the contributions to the calibration objective function are sensitive to different model parameters. For instance, the climatology constrains the Nash cascade parameters, while the FDC performance statistic is not very sensitive to changes in those parameters. The parameter sensitivity indices relative to the GRACE objective are more evenly distributed. The altimetry objective is most sensitive to the routing parameters, in particular channel roughness. Comparison between the calibration objective and the contributions shows a clear dominance
of the FDC in the aggregated objective function. Simulating the full Pareto front allows the user to assess trade-offs between individual contributions but is computationally expensive.


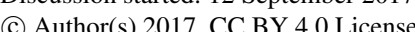


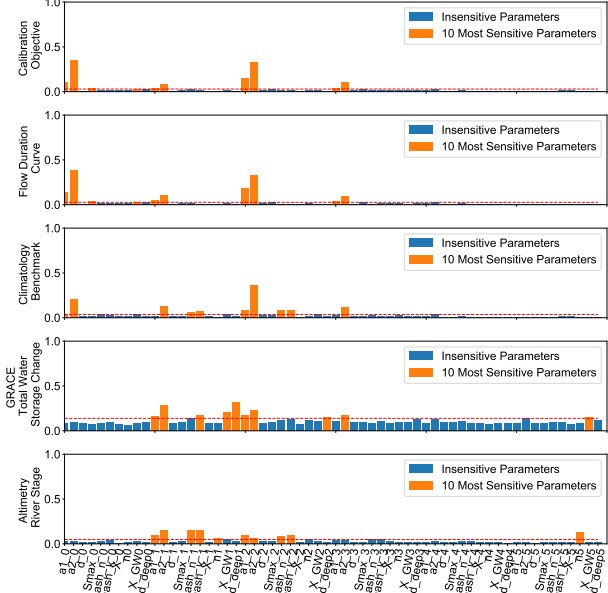

**Figure 6.** Sensitivity analysis of the model parameters on contributions to aggregated objective function (top) on (from second row down): FDC, climatology, GRACE and altimetry water height.

The aggregated objective function values are 0.81 and 0.86 for TRMM and FEWS respectively. The models perform very similarly regardless of the climate forcing, although the statistics of the TRMM model are slightly better overall. Evaluation of the parameter space post-calibration in Figure 7 shows a clear convergence of all parameters to their optimal value. Only X appears to be less constrained in the shown region.





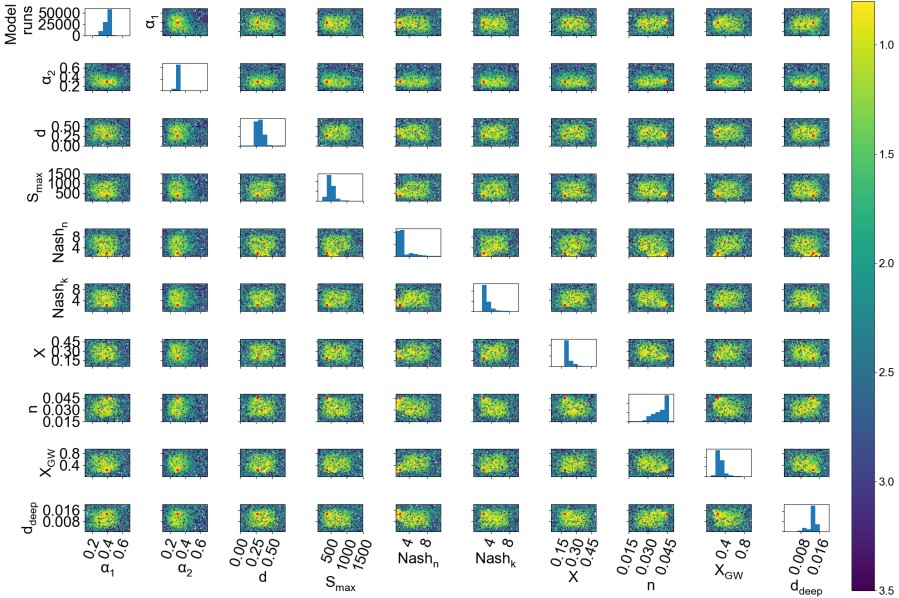

**Figure 7.** Parameter space post-calibration for the Ngounié calibration zone and the TRMM model. The yellow dots represent the best model runs and the red dots indicate the best parameter values. Example for TRMM model, Ngounié parameters.

All calibrated parameter values of both models are provided in Table 3 Very few parameters converged to the upper boundary of the a-priori parameter interval: $k_{Nash}$ is close but not equal to the lower boundary in the TRMM model in the Ogooué Delta and Ndjolé region, and $X_{GW}$ is equal to 0.95 in the FEWS-RFE model in the Eastern Gabon Plateau region. The a-priori parameter interval could be extended to allow larger values of $X_{GW}$, and consequently close to no recharge to the deep aquifer.

5   Parameter correlation between the parameters governing the partitioning of water between different reservoirs and delaying runoff is inevitable. Both parameter sets are physically reasonable and the basin median is very similar between the two models; however, some of the most sensitive parameters are quite different, suggesting a propagation of the difference in precipitation through the model. In particular, the TRMM parameters are more heterogeneous throughout the basin. Furthermore, the TRMM model has a higher retention efficiency in 4 out of 6 regions and a higher ET efficiency in all basins (larger $\alpha$ values). The

10  TRMM model also has more recharge to the deep aquifer (smaller $X_{GW}$ in all regions). This reflects that TRMM predicts larger volumes of precipitation throughout the basin.



**Table 3.** Calibrated parameters from the two models forced by TRMM and FEWS-RFE precipitation. The numbers in parenthesis correspond to the subscript indices in Figure 6.

|  | $\alpha_1$ | $\alpha_2$ | d | $S_{max}$ | $n_{Nash}$ | $k_{Nash}$ | $X_{GW}$ | $d_{deep}$ | X | n |
|---|---|---|---|---|---|---|---|---|---|---|
| **Batéké Plateau (2)** | | | | | | | | | | |
| TRMM | 0.41 | 0.25 | 0.18 | 466 | 5 | 1.17 | 13.4 | 0.010 | 0.22 | 0.019 |
| FEWS-RFE | 0.38 | 0.23 | 0.26 | 633 | 5 | 3.25 | 25.0 | 0.006 | 0.22 | 0.017 |
| **Eastern Gabon Plateau (0)** | | | | | | | | | | |
| TRMM | 0.53 | 0.30 | 0.31 | 559 | 6 | 4.43 | 87.6 | 0.016 | 0.29 | 0.026 |
| FEWS-RFE | 0.64 | 0.27 | 0.19 | 795 | 7 | 3.34 | 95.0 | 0.014 | 0.13 | 0.037 |
| **Booué (3)** | | | | | | | | | | |
| TRMM | 0.57 | 0.28 | 0.42 | 844 | 5 | 5.67 | 3.4 | 0.018 | 0.36 | 0.041 |
| FEWS-RFE | 0.43 | 0.26 | 0.47 | 934 | 4 | 3.88 | 19.4 | 0.015 | 0.42 | 0.050 |
| **Ndjolé (4)** | | | | | | | | | | |
| TRMM | 0.26 | 0.61 | 0.30 | 1142 | 6 | 0.20 | 36.4 | 0.008 | 0.22 | 0.049 |
| FEWS-RFE | 0.24 | 0.53 | 0.43 | 737 | 4 | 4.84 | 38.3 | 0.015 | 0.38 | 0.015 |
| **Ngounié (1)** | | | | | | | | | | |
| TRMM | 0.39 | 0.30 | 0.27 | 379 | 2 | 2.29 | 24.9 | 0.013 | 0.19 | 0.044 |
| FEWS-RFE | 0.44 | 0.27 | 0.21 | 1524 | 1 | 5.69 | 30.8 | 0.013 | 0.26 | 0.040 |
| **Ogooué Delta (5)** | | | | | | | | | | |
| TRMM | 0.42 | 0.20 | 0.47 | 856 | 5 | 0.44 | 9.5 | 0.011 | 0.14 | 0.036 |
| FEWS-RFE | 0.24 | 0.20 | 0.64 | 797 | 6 | 5.05 | 52.0 | 0.018 | 0.24 | 0.034 |
| **Basin median** | | | | | | | | | | |
| TRMM | 0.41 | 0.28 | 0.31 | 702 | 5 | 3.35 | 19.15 | 0.012 | 0.22 | 0.036 |
| FEWS-RFE | 0.38 | 0.25 | 0.35 | 796 | 5 | 4.36 | 34.55 | 0.015 | 0.25 | 0.034 |

## 4.2 Spatial Characterization of discharge

Figure 8 shows the observed and simulated flow duration curves (FDC) and climatology at the downstream calibration station, Lambaréné. The flow regime in the Ogooué consists of precipitation-driven direct runoff peaks as seen from the steep slope of the FDC for low exceedance probabilities, and a sizeable baseflow, characterized by a non-zero minimum flow value and a flattening curve at higher exceedance probabilities. Generally, the FDC simulated by both models are within 10% of the




observed FDC at all six calibration stations ($R_{FDC} \geq 0$). Furthermore, the calibration is deemed reasonable if the simulated climatology falls within one standard deviation of the observation (WRMSD $\leq 1$). For both models, this is the case at all calibration stations.

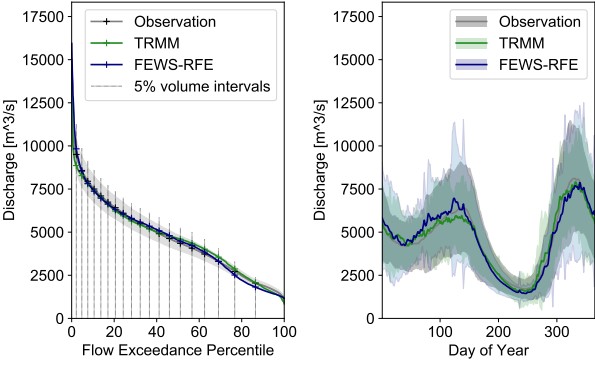

**Figure 8.** Flow duration curves and daily discharge climatology benchmark at the Lambaréné calibration stations, the surfaces in the climatology plot represent the 90% confidence interval.

Table 4 shows the performance statistics for the FDC and climatology contributions to the calibration objective at the calibration and validation stations. Both models are within the validation criteria at all calibration stations and two out of five validation stations. Overall, the performances of the two models are similar in terms of simulating flow regime in the basin: The TRMM model performs better based on 10 out of 19 validated performance measures.

The calibration objective incorporates two important evaluation criteria: the model's ability to capture the seasonality and probability distributions of discharge throughout the basin. The results indicate the model is capable of simulating both, regardless of precipitation forcing. Day-to-day comparison with up-to-date discharge is necessary to evaluate the success of the calibration strategy compared to traditional approaches but in cases where no current observations are available, the approach used in this study is a good compromise.





**Table 4.** Performance measures for the TRMM and FEWS-RFE models based on the discharge observations. The number between parenthesis is the location of the discharge stations in the model subbasins. Values in bold highlight the best validated performance.

| | Station (reach) | $R_{FDC}$ | | Climatology, WRMSD | |
|---|---|---|---|---|---|
| | | TRMM | FEWS-RFE | TRMM | FEWS-RFE |
| **Batéké Plateau** | | | | | |
| Calibration | Lastoursville | 0.39 | **0.63** | 0.56 | **0.33** |
| Validation | Leyami | -0.08 | -0.14 | 1.43 | 1.14 |
| **Eastern Gabon Plateau** | | | | | |
| Calibration | Makokou | **0.43** | 0.36 | 0.65 | **0.59** |
| Validation | Belinga | -0.64 | -0.76 | **0.68** | 0.92 |
| **Booué** | | | | | |
| Calibration | Booué | 0.60 | **0.79** | **0.41** | 0.51 |
| Validation | Loa-Loa | -0.05 | -0.04 | **0.62** | 0.75 |
| **Ndjolé** | | | | | |
| Calibration | Ndjolé | **0.67** | 0.60 | **0.52** | 0.82 |
| Validation | Portes de l'Okanda | 0.58 | **0.71** | **0.38** | 0.58 |
| **Ngounié** | | | | | |
| Calibration | Fougamou | **0.76** | 0.67 | 0.37 | **0.36** |
| Validation | Sindara | **0.68** | 0.67 | 0.57 | **0.53** |
| **Lower Ogooué** | | | | | |
| Calibration | Lambaréné | 0.67 | **0.71** | **0.31** | 0.42 |

## 4.3 Simulated Total Water Storage Change

Figure 9 shows the total water storage change observed by GRACE and simulated by the two models in the two basin halves. The monthly total water storage simulated by the model consists of the sum of water stored in the root zone, the shallow aquifer and deep aquifer, the tributary processes and the main channel. The tributary processes represent 9.2% and 10.1% of the total storage change throughout the basin in the TRMM and FEWS-RFE model respectively, indicating a significant contribution from water retention processes. This is consistent with the large number of wetlands and lakes in the basin. The deep aquifer holds the lion's share in total water storage change: respectively 70.6% and 83.0% of total water storage change in the TRMM and FEWS-RFE models, while the soil storage contributes 12.2% and 1.9% and around 2.5% of the change originates from the shallow aquifer in both models. Storage changes in the main channel contribute 5.5% and 2.4% to the total water storage change respectively. The largest difference between the models relates to the changes in soil water storage. The



TRMM model generally has larger  parameters: larger retention efficiency leads to larger positive soil storage changes and higher evapotranspiration efficiency leads to larger negative soil storage changes.

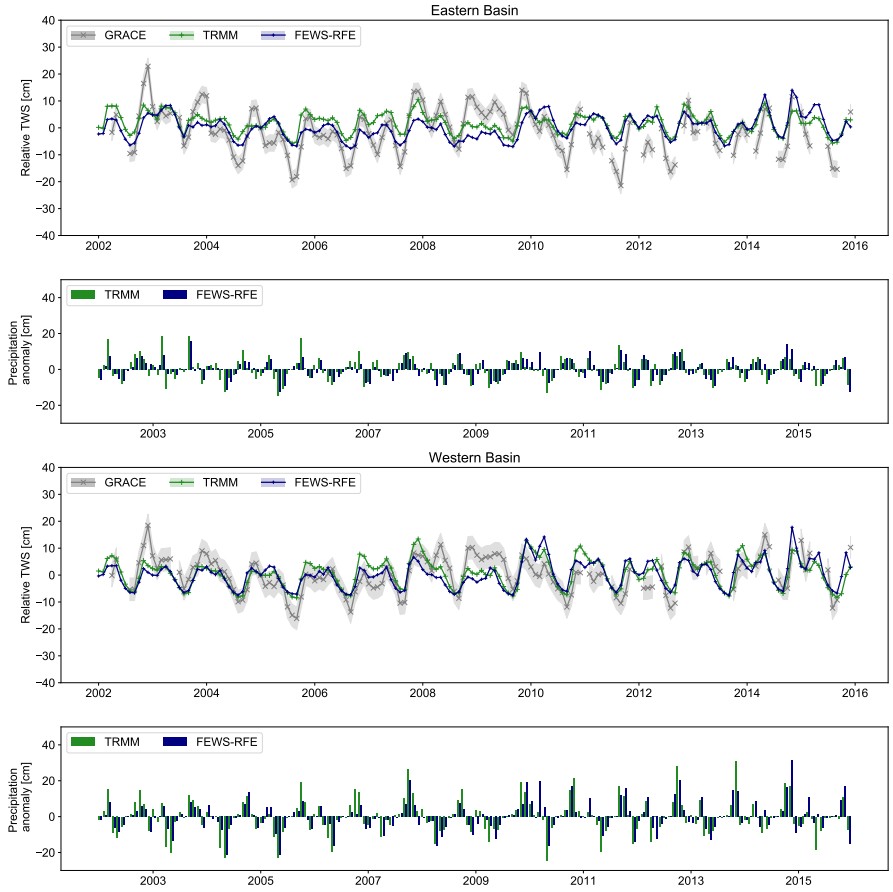

**Figure 9.** Total water storage change and precipitation anomaly referenced to the monthly climatology over the period of simulation for the Eastern (top) and Western (bottom) basins.

Table 5 shows the performance statistics for the TWSC contribution. The TRMM model generally performs better although the performance statistics are higher than the validation criteria (WRMSE $\leq$ 1), suggesting the residuals exceed the obser-
vation uncertainty. However, the models both capture the TWSC in the basin quite well, albeit storage change is generally underestimated. The best performance is achieved in the Western basin (bottom plots), with WRMSE values below 1.4 for the calibration and validation period. We compute the precipitation anomalies relative to the mean monthly precipitation. On average, the TRMM estimates fluctuate more, as seen in the larger anomalies (5.8 cm per month, compared to 5.0 cm for FEWS-RFE). Due to the delay between precipitation signal and storage response, we obtain better fits in years where the pre-
cipitation anomalies match the observed storage change: e.g. in late 2006-early 2007, FEWS-RFE estimated more rain during the rainy season, resulting in an overestimation of the relative TWS in the subsequent year. Similarly, both products predict





little to no positive water storage change in 2009 and have a larger number of negative than positive precipitation anomalies. Thus, the discrepancies between the GRACE observations and the simulated total water storage changes can be attributed to three factors: the trade-off between fitting the water storage in the basin versus other calibration objectives, uncertainties in the GRACE observations, particularly considering the size of the study region and the spatial resolution of the observations and

finally, differences in trends in water storage and precipitation anomalies. The latter can be due to water retention or diversion in the basin not accounted for by the model or to uncertainties in the precipitation estimates.

**Table 5.** GRACE objective functions for the two models for the calibration and validation periods.

|  | Calibration | | Validation | |
|---|---|---|---|---|
|  | TRMM | FEWS-RFE | TRMM | FEWS-RFE |
| East | 2.11 | 2.19 | 2.55 | 2.68 |
| West | 1.21 | 1.33 | 1.16 | 1.33 |

## 4.4 River Stage

For comparative purposes, we reference the observed and simulated water heights to the long term mean. At virtual stations, we calculate the long-term mean based only on dates where satellite observations were acquired. The results are shown in Table

6. Simulated water depths depend on the river cross-sectional geometry. We do not calibrate river cross-sectional geometry in order to limit the number of fitting parameters. Nevertheless, the simulated depth amplitudes are realistic. The simulated amplitudes are within the 90% confidence intervals of the observation at all but one virtual station. The NSE is above 0.5 during the calibration period in nine out of 12 virtual stations for the TRMM forced model and in eight out of 12 for the FEWS-RFE model. Performance slightly decreases in the validation period, in particular for the Ngounié virtual stations and

the FEWS-RFE model. When comparing the simulated water depth amplitudes, to those observed at each station, the RMSD is 0.74 m for the TRMM model and 0.87 m for the FEWS-RFE, corresponding to 0.85 and 0.94 times the standard deviation of annual water height amplitude (Table 7).This is comparable to the study by Schneider et al. (2017), in which they obtained an average RMSE of 0.83 m for the Brahmaputra after calibrating the river cross-sections in a hydrodynamic model against Envisat virtual stations. Figure 10 shows the water height fluctuations at two of the virtual stations.





**Table 6.** Performance statistics for altimetry at Virtual Stations – for the individual virtual stations, the first line shows the statistics for the calibration period, the second for the validation period. Values in bold are within the validation criteria (NSE > 0.5, WRMSD ≤ 1).

| Virtual Station Mission (Subbasin ID) Coordinates, chainage | Amplitude [m] Altimetry Mission [90% CI] | TRMM | FEWS | NSE TRMM | NSE FEWS | WRMSD TRMM | WRMSD FEWS |
|---|---|---|---|---|---|---|---|
| **Ogooué** | | | | | | | |
| Envisat (12) | 2.35 [1.52-3.87] | 1.21 | 1.08 | 0.43 | 0.20 | **0.81** | **0.96** |
| 1.224°S, 13.334°E, 695 km | | | | **0.61** | **0.51** | **0.70** | **0.79** |
| Envisat (20) | 4.22 [1.17-5.39] | 2.83 | 2.02 | **0.60** | 0.41 | 1.54 | 1.86 |
| 0.061°S, 11.642°E, 385 km | | | | 0.21 | 0.10 | 1.70 | 1.81 |
| Envisat (24) | 2.87 [1.25-3.65] | 1.80 | 1.82 | **0.74** | **0.63** | **0.72** | **0.86** |
| 0.506°S, 10.302°E, 187 km | | | | 0.46 | 0.25 | **1.00** | 1.18 |
| Envisat (26) | 2.87 [1.84-4.71] | 2.70 | 2.72 | **0.78** | **0.77** | **0.68** | **0.70** |
| 0.835°S, 10.027°E, 133 km | | | | **0.53** | -0.08 | **0.95** | 1.44 |
| Envisat (26) | 3.74 [2.06-5.80] | 3.40 | 3.71 | **0.67** | **0.73** | 1.14 | 1.04 |
| 0.921°S, 9.675°E, 83 km | | | | **0.52** | -0.33 | 1.30 | 2.15 |
| Envisat (27) | 2.42 [1.54-3.96] | 2.22 | 2.27 | **0.78** | **0.75** | **0.57** | **0.61** |
| 1.073°S, 9.256°E, 30 km | | | | **0.55** | 0.15 | **0.90** | 1.30 |
| **Ivindo** | | | | | | | |
| Jason-2 (10) | 4.72 [1.13-5.85] | 3.80 | 4.15 | 0.34 | 0.33 | 2.00 | 2.02 |
| 1.1°N, 13.076°E, 677 km | | | | 0.06 | -0.12 | 2.13 | 2.32 |
| Envisat (14) | 2.22 [1.11-3.33] | 1.56 | 1.74 | **0.62** | **0.57** | **0.75** | **0.80** |
| 0.251°N, 12.422°E, 533 km | | | | 0.39 | 0.11 | **0.83** | **1.00** |
| **Ngounié** | | | | | | | |
| Envisat (7) | 2.69 [1.44-4.13] | 3.74 | 2.79 | **0.66** | **0.64** | **0.87** | **0.89** |
| 1.272°S, 10.650°E, 305 km | | | | -0.48 | -1.72 | 1.94 | 2.63 |
| Envisat (11) | 2.42 [1.43-3.86] | 2.65 | 2.67 | **0.82** | **0.73** | **0.54** | **0.67** |
| 1.142°S, 10.678°E, 273 km | | | | 0.05 | -0.59 | 1.38 | 1.78 |
| Envisat (11) | 3.18 [1.19-4.37] | 2.86 | 2.77 | 0.41 | 0.39 | 1.43 | 1.46 |
| 1.042°S, 10.701°E, 263 km | | | | -0.24 | -0.68 | 1.57 | 1.83 |
| Envisat (15) | 2.99 [2.04-5.03] | 2.84 | 2.63 | **0.75** | **0.55** | **0.73** | **0.97** |
| 0.601°S, 10.323°E, 183 km | | | | 0.39 | -0.39 | 1.22 | 1.83 |





**Table 7.** Basin amplitude statistics at all virtual stations. The percentages are relative to the mean observed amplitude.

| WRMSE | | RMSD [m] (%) | | MD [m] (%) | |
|---|---|---|---|---|---|
| TRMM | FEWS | TRMM | FEWS | TRMM | FEWS |
| 0.85 | 0.94 | 0.74 | 0.87 | 0.41 | 0.42 |
| | | (24.8%) | (28.8%) | (13.8%) | (14.0%) |

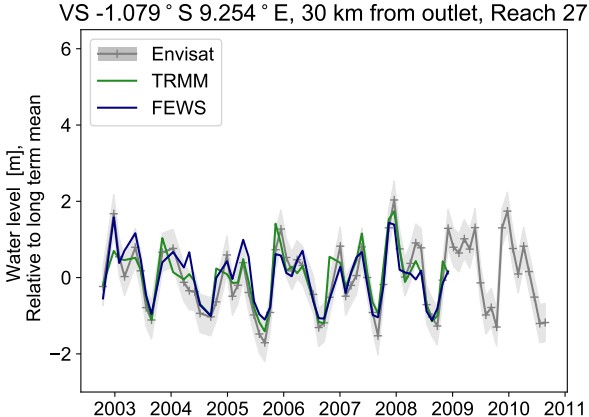

**Figure 10.** Results for water height simulation at a selected virtual stations.

Figure 11 shows the simulated water height anomaly climatology from the Batéké Plateau to the Delta, and all available altimetry observations. Sharp changes in amplitude reflect the confluence of river branches briefly increasing width (e.g. chainage 450-420 at the confluence of the Ivindo and the Ogooué) and to the nature of the topography: the river is narrow between Booué and Ndjolé (chainage 420-260), before reaching the plain and eventually the delta, where the river width reaches up to 1300

m. At chainage 180, the Ngounié joins the Ogooué and the river width increases by 500 m. The temporal pattern agrees well and the spatial patterns are comparable. The RMSD between CryoSat-2 anomalies and model simulations between 1 and 2 m in most regions in the basin (Table 8), part of which can be attributed to the approximated mean water level and to the time of observation. Due to its long repeat period, CryoSat-2 samples more often during certain seasons over different parts of the river. Schneider et al. (2017) obtained an RMSD of 2.5 m between simulated water heights and CryoSat-2 observations over

the Brahmaputra – thus we deem the obtained results satisfactory in light of the available information.





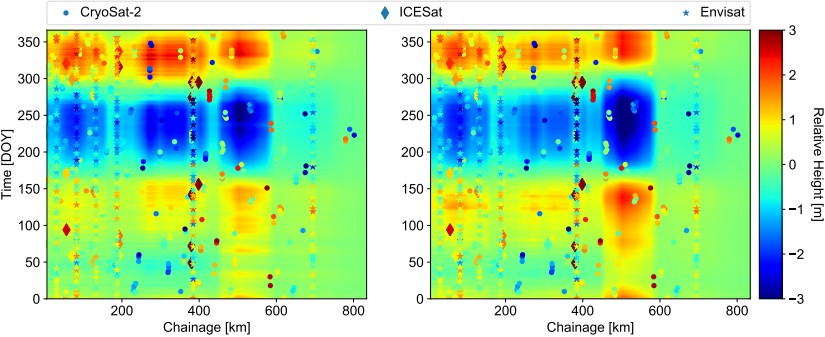

**Figure 11.** Interpolated relative water heights [m] based on the simulated daily water depth climatology in the Ogooué by the model forced with TRMM 3B42 v7 (top) and the FEWS-RFE (bottom) precipitation and altimetry observations from CryoSat-2, ICESat and Envisat. The VS visible in the figure correspond to those in reach 27, 26, 24, 20 and 12.

**Table 8.** CryoSat-2 versus simulated relative water depths.

| River stretch | Number of CryoSat-2 observations | RMSD [m] | | Bias [m] | |
|---|---|---|---|---|---|
| | | TRMM | FEWS-RFE | TRMM | FEWS-RFE |
| Upstream of Makokou (Ivindo) | 32 | 1.76 | 1.69 | 0.01 | 0.07 |
| Upstream of Sindara (Ngounié) | 47 | 2.37 | 2.06 | -0.19 | 0.10 |
| Between Ndjolé and Lambaréné (Ogooué) | 46 | 0.94 | 0.98 | 0.02 | -0.02 |
| Downstream of Lambaréné (Ogooué) | 110 | 1.03 | 1.15 | -0.08 | -0.14 |
| Ogooué river | 353 | 1.85 | 1.85 | -0.09 | -0.21 |

While altimetry observations from drifting ground track missions increase the spatial resolution, observations from the virtual stations give a temporal characterization of water height fluctuations at specific locations in the basin. The obtained accuracy is of the order of magnitude of values reported in the literature – better results could be obtained with knowledge about the bathymetry or by calibrating the river cross-sections. In this study, increasing the number of calibration parameters would not be suitable because only a limited number of CryoSat-2 observations are available and no contemporary discharge observations to validate timing.

Similarly, to the water storage amplitudes, the water level amplitudes are slightly underestimated, particularly in the Eastern basin. The model parameters are most sensitive to improving the FDC and climatology benchmark contributions, which are based on historical discharge observations. Changes in precipitation patterns since the time of observation are likely to have affected discharge patterns. The comparison to contemporary satellite altimetry observations strengthens the validation of the model; however, the underestimation of water height amplitude and total water storage change in the basin may indicate that





the model compensates for changes in precipitation patterns and uncertainties in the precipitation products in order to fit the historical discharge dataset.

## 4.5 Discussion

This study uses free, publicly available remote sensing observations relevant to the proposed model structure to characterise the basin. Several more types of remote sensing products are available but not included. For instance, no reliable soil moisture estimates can be produced for the Ogooué basin because the dense vegetation masks the microwave returns from the underlying soil (Tang et al., 2009). We select the most relevant products and explore how new data sources may supplement existing datasets and extend their applicability. To the authors' knowledge, this study is the first to use SAR imagery from Sentinel-1 to extract CryoSat-2 observations over an inland water body, and the first study to evaluate CryoSat-2 observations over the Ogooué. The size of the Ogooué (approximately 1.3 km at its widest and 390 m on average), makes it as an interesting study area for altimetry observations. However, without cloud penetrating technologies, it would be very difficult to produce a satisfactory water mask of the river. The possibility to develop detailed water masks for virtually any inland water body from SAR imagery greatly expands the applicability of altimetry observations from drifting ground track missions over rivers.

In poorly gauged basins, the paucity of observations limits the estimation of the model parameters and consequently model complexity (Johnston and Smakhtin, 2014). Remote sensing data has been used in several studies to compensate for gaps in in-situ observations and has enabled the definition of distributed or semi-distributed models even in poorly gauged basins (van Griensven et al., 2012). Furthermore, the accessibility of remote sensing observations creates modelling opportunities in basins, where in-situ data is insufficient on its own (Johnston and Smakhtin, 2014). This is the case for the Ogooué, which to the author's knowledge only has decade-old discharge observations and precipitation records at a dozen stations. The model used in this study has a fairly simple and flexible structure with few parameters and limited input data requirements, which can accommodate several basin and river network configurations. Furthermore, although the model currently does not support reservoir characterization or abstraction losses, these can be implemented within the model structure. By starting with a simple structure and gradually adding complexity (deep aquifer, tributary processes), the principle of parsimony is respected.

The remote sensing observations used in this study help characterize the otherwise ungauged basin and the model can produce valuable information for water managers. Several studies have benefitted from including altimetry observations (Schneider et al., 2017; Michailovsky et al., 2013; Schumann and Domeneghetti, 2016) and total water storage observations (Xie et al., 2012; Milzow et al., 2011) in river basin models. In this study, the altimetry missions generally agree very well and the observations provide valuable information on water heights throughout the river. Although it would be useful to confirm the remotely sensed observations with ground observations, the availability of contemporary observations strengthened the evaluation of the hydrological model of the Ogooué. Without additional observations or on-ground information on the basin, the presented model is the best available representation of the Ogooué basin. However, model simulations can never replace observations and remote sensing observations have never been evaluated in the basin before. Therefore, ground truthing efforts and in-situ gauging campaigns would greatly strengthen the conclusions of this study.

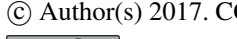



A model should always be evaluated in light of its intended application (Johnston and Smakhtin, 2014). The model developed in this study, is the first model of the Ogooué river basin, and provides otherwise unavailable information regarding the baseline river flow regime. It can be used in a broad range of applications, including flood forecasting, climate change evaluation and as an impact assessment tool for planned water infrastructure investments. For instance, the hydrologic impact of hydraulic 5 infrastructures at the inlet to downstream key wetlands resolved by the model can be assessed and compared to the baseline developed in this study.

## 5 Conclusions

In this study, we explore the use of multi-mission remote sensing to inform a hydrological model of the fourth largest African river by discharge, the Ogooué in Gabon. We set up a lumped conceptual rainfall-runoff model based on the Budyko framework 10 coupled to a Muskingum routing scheme. We force the model using remote sensing precipitation and calibrated using a combination of historical in-situ discharge observations from the 1960s and 1970s, and total water storage observations from the GRACE mission. Remote sensing enables the evaluation of the model against contemporary observations and helps constrain model parameters by including information other than discharge measurements.

In addition, this study shows the potential of the new ESA Sentinel missions by deriving a detailed river mask from Sentinel-
1 radar imagery, used to extract altimetry observations from CryoSat-2. The multi-mission approach increases spatial and temporal coverage and acts as a useful supplement to the observed in-situ discharge in terms of validation, in regions were the missions agree. We validate the water height simulations against the altimetry observations at multiple points in the basin. With the methods applied in this study, a dynamic river mask can be defined and used to extract relevant observations over inland water bodies of interest from existing and new satellite altimetry missions. New radar altimetry missions as Sentinel-3
carrying state-of-the-art equipment are expected to provide higher accuracy observations. Combined with the water masking method proposed in this study, relevant time series of river water heights can be extracted and used in hydrological modelling studies.

Progress in remote sensing technologies, instruments and extraction algorithms now allows for the observation of most hydrological states and fluxes from space. This offers a unique possibility to obtain observations in poorly gauged or remote areas
and to supplement hydrological modelling applications with the necessary input-data and useful observations for parameter estimation. The model used in this study can be applied in scenario evaluations and provides an otherwise unavailable insight into the hydrological regime of the Ogooué at catchment scale. By combining hydrological modelling with multi-mission remote sensing from ten different satellite missions, we obtain new information on an otherwise unstudied basin. The proposed model is the best current baseline characterization of hydrological conditions in the Ogooué in light of the available observations.

*Code and data availability.* The python code used in this study will be publicly available in upcoming versions of the GlobWetlands-Africa Toolbox. All data sets used in this study are derived from publicly available resources.





*Competing interests.* The authors declare that they have no conflict of interest

*Acknowledgements.* We acknowledge funding from the European Space Agency (ESA) through the GlobWetlands-Africa project. Daily historical in situ observations of discharge in the river basin recorded by the Office de la Recherche Scientifique et Technique Outre-Mer (ORSTOM) and the Direction Générale des Ressources Hydrauliques (DGRH) in Gabon were recovered from the Institut de Recherche

5    pour le Développement (IRD) and are accessible on their Système d'Informations Environnementales sur les Ressources en Eaux et leur Modélisation (SIEREM) website, http://www.hydrosciences.fr/sierem.





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
