# Peer review of "Informing a hydrological model of the Ogooué with multi-mission remote sensing data"

_Hydrology and Earth System Sciences, 2017_

## Referee Comment (RC1) · Anonymous Referee #1 · 27 Sep 2017

This paper is of a great interest to the community of hydrologists in Africa. It is a first attempt to estimate the seasonnal river discharge and its interrannual variability of the Ogooue River in Gabon, from satellite data only. The main interest of this project is to build simulated water heights and discharge time series for virtual gauging stations along the river course, while discharge observed time series end in 1984 for most of the stations, and rainfall data are also difficult to update. The satellite data and the methods used are validated against some in situ data series, and show a good capacity to simulate coherent discharge time series for most of the stations, even if the absolute precision remain of several tenth of cm, which is still difficult to use for real time operational alerts. The paper is well organised and written, and the illustrations are appropriate. I only recommend some minor corrections to clarify some points,

enlarge the references list to a few uncited papers related to the core of the study (at least Mahe et al. 1990, see below), and correct some minor errors. Detailed remarks. Some paragraphs are written in bold, to modify. P1 Lines 16-17: the abstract indicates that this study is the best current baseline characterization of hydrological conditions in the Ogooué river. It is partly true, if you consider the previous publication of Mahe et al. 2013 which shows monthly discharges for the Ogooue river over the period 2000-2007 in regard of previous periods until 1990 (the 90's are missing time series).  P 2 Line 27: accuracy between 30 and 70 cm: can the author estimate the discharge value error considering this height margin?  P 3 Lines 25-29: About the previously used models, lumped models have proved less efficient to represent the two annual flood peaks of equatorial rivers, mainly due to a very approximative estimation of PE (Paturel et al 2003) (Dezetter et al 2008) Paturel, J.E., Ouedraogo, M., Mahe, G., Servat, E., Dezetter, A., Ardoin, S. (2003).  The influence of distributed input data on the hydrological modelling of monthly river flow regimes in West Africa.  Hydrological Sciences Journal, 48, 6, 881-890.  Dezetter, A., Girard, S., Paturel, J.E., Mahé, G., Ardoin-Bardin, S., Servat, E. (2008).  Simulation of runoff in West Africa: Is there a single data-model combination that produces the best simulation results ?  Journal of Hydrology, 354, 203-212.

P 4 Line 29: Hydrological monitoring efforts "by ORSTOM hydrologists during the 50's to the 80's" P 4 Line 30: ". . .available informations are from 1984" for most stations, (Mahe et al., 1990; 1994) Mahé, G., Lerique, J., Olivry, J.C. (1990).  L'Ogooué au Gabon.  Reconstitution des débits manquants et mise en évidence de variations climatiques à l'équateur. Hydrologie Continentale, Ed. ORSTOM, Paris, 5, 2, 105-124. Mahé, G., Delclaux, F., Crespy, A. (1994).  Elaboration d'une chaîne de traitement pluviométrique et application au calcul automatique de lames précipitées (bassin-versant de l'Ogooué au Gabon). Hydrologie Continentale, 9, 2, 169-180. P 4 Libne 32: there is much more in the paper of Mahe et al. 2013 (update of the 1990's paper), for instance the dramatic reduction of the Spring flood at Lambarene since the 80's, confirmed during the 2000's as showed in the 2013's paper. P 5 Figure 1: the text in

white is difficult to read P 6 Line 7: historical precipitations at four locations: which ones? P 9 Figure 3: too small P 11 Line 1: the storage constants are fixed how? And at which value? P 11: 3.7 Watershed Delination: Why not used the existing delineation available at the SIEREM website? This site is cited by the authors, but it is difficult to know for what purpose it is cited. http://www.hydrosciences.fr/sierem/index_en.htm http://www.hydrosciences.fr/sierem/consultation/consultationgraphbas.asp?basid=OGOOUE http://www.hydrosciences.fr/sierem/produits/gis/Ogooue.asp free GIS files soil WHC for $\frac{1}{2}$ square degrees, from the FAO soil map of the world. Gives the water height for the upper soil layer. Please cite Boyer et al., 2006 to refer to SIEREM Boyer, J.F., Dieulin, C., Rouché, N., Crès, A., Servat, E., Paturel, J.E., Mahé, G. (2006). SIEREM: an environmental information system for water resources. In: Water Resource Variability: Hydrological Impacts. Proc. of the 5th FRIEND World Conference, La Havana, Cuba, IAHS Publ. 308, 19-25. P 19 Table 4: the capyion mentions number between parenthesis, but there are none in the table. Please clarify. P 19 Line 7: total water storage 70.6 and 83%. OK, but which part of this pcentage participates to surface runoff? P 25 Line18-19: there are more than a few decades of observations for the Ogooue river at Lambarene, the time series starts in 1929, and some missing years have been reconstructed. See Mahe et al. 1990 P 27 Line 6: OK to thank SIEREM, but the authors should refere to the Boyer et al 2006 paper (see up)

Please also note the supplement to this comment:
https://www.hydrol-earth-syst-sci-discuss.net/hess-2017-549/hess-2017-549-RC1-supplement.pdf

---

## Referee Comment (RC2) · Anonymous Referee #2 · 13 Oct 2017

The paper investigated the use of multi-mission remote sensing data to force, calibrate and validate a lumped conceptual rainfall-runoff model on an ungauged Ogooué river basin in Africa. The paper is clear and well written. I enjoyed to read this study because is well thought out and organized. The Figures are appropriated even if I would prefer bigger (especially Figure 3, 6 and 7). I recommend the publication of the paper after minor changes below specified.

Some references are not properly assigned to the concept. An example is the paper of Berry et al. (2012) mentioned at P1 Line24 and P2 Line 13 to underline the decline of in-situ gauging networks. I think different papers can replace this citation [1, 2, 3]. Please check also the reference Schumann and Domeneghetti (2016) at P25 Line 26.

Plots a, b, c of Figure 2 are not mentioned and commented in the text. Please provide

description. Moreover, P6 Line 4 "(Figure 2, c and d)" should be replaced with "(Figure 2, d and e)".

P19 Line 2: "... and simulated the two models in the two basin helves". What does the authors mean with "two models"?

Table 4: in the caption parenthesis are mentioned but they are not present in the table. Please correct.

Table 7: the acronym MD is not specified in the text or in the caption.

References:

[1] C. Vörösmarty, A. Askew, W. Grabs, R. G. Barry, C. Birkett, P. Döll, B. Goodison, A. Hall, R. Jenne, L. Kitaev, J. Landwehr, M. Keeler, G. Leavesley, J. Schaake, K. Strzepek, S. S. Sundarvel, K. Takeuchi and F. Webster, "Global water data: A newly endangered species," Eos Trans AGU, vol. 82, no. 5, pp. 54–58, Jan. 2001.

[2] N. Sneeuw, C. Lorenz, B. Devaraju, M. J. Tourian, J. Riegger, H. Kunstmann and A. Bárdossy, "Estimating runoff using hydro-geodetic approaches," Surv. Geophys., vol. 35, no. 6, pp. 1333–1359, 2014.

[3] D. M. Hannah, S. Demuth, H. A. J. van Lanen, U. Looser, C. Prudhomme, G. Rees, K. Stahl and L. M., Tallaksen, "Large‐scale river flow archives: importance, current status and future needs," Hydrol Process., vol. 25, no. 7, pp. 1191–1200, Mar. 2011.

---

## Author Comment (AC1) · 16 Oct 2017

Response to the review of Anonymous Referee #1, posted 27/09/2017

This paper is of a great interest to the community of hydrologists in Africa. It is a first attempt to estimate the seasonnal river discharge and its interrannual variability of the Ogooue River in Gabon, from satellite data only. The main interest of this project is to build simulated water heights and discharge time series for virtual gauging stations along the river course, while discharge observed time series end in 1984 for most of the stations, and rainfall data are also difficult to update. The satellite data and the methods used are validated against some in situ data series, and show a good capacity to simulate coherent discharge time series for most of the stations, even if the

absolute precision remain of several tenth of cm, which is still difficult to use for real time operational alerts. The paper is well organised and written, and the illustrations are appropriate. I only recommend some minor corrections to clarify some points, enlarge the references list to a few uncited papers related to the core of the study (at least Mahe et al. 1990, see below), and correct some minor errors.

REPLY: We thank the referee for the feedback and comments on the article. Particularly, we thank the referee for the suggested references, and for providing the paper by Mahe et al. (1999). Although we were aware of the paper through citations, we had not been able to retrieve it or the paper by Mahe et al. (1994). We will refer to the paper by Mahe et al. from 1990 in the article. Would it be possible for the referee to share the 1994 paper as well?

P1 Lines 16-17: the abstract indicates that this study is the best current baseline characterization of hydrological conditions in the Ogooué river. It is partly true, if you consider the previous publication of Mahe et al. 2013 which shows monthly discharges for the Ogooue river over the period 2000- 2007 in regard of previous periods until 1990 (the 90's are missing time series).

REPLY: Indeed the previous publication by Mahe et al. (2013) also offers a characterization of hydrological conditions based on the most up-to-date in-situ observations of discharge at the Lambaréné station. The current study is the first example of a basin-scale representation of the Ogooué river regime, including fluxes and storages at a daily time step, which can serve as a stepping-stone for simulations of scenarios of change in the basin.

P 2 Line 27: accuracy between 30 and 70 cm: can the author estimate the discharge value error considering this height margin?

REPLY: Estimating river discharge from radar altimetry observations (and thus the propagation of uncertainty into discharge values) is quite tricky, as it requires information on river bathymetry and the establishment of a rating curve. Michailovsky et al.

[Figure]

(2012) converted radar altimetry observations to discharge using in-situ observations from field campaigns and historical records and obtained RMSE values ranging from 4.5 to 7.2% of the mean annual discharge amplitude, corresponding to between 19.9 and 69.4 mˆ3/s for a water level RMSE between 30 and 70 cm relative to the in-situ levels. While this is possible for missions with repeat ground tracks crossing the river line at specific points with relatively short return periods (e.g. 30 days for Envisat), bathymetry observations throughout the entire river would be required in order to apply this to the CryoSat-2 mission, which is highly impractical.

Furthermore, as historical rating curves and bathymetry observations are not available for the Ogooué, we compare the altimetry water height amplitudes observed by radar altimetry directly to the simulated water height amplitudes. Thus we do not need to estimate discharge. We use Envisat and Jason-2 observations in the calibration and the accuracy of these missions do impact the estimated discharge values, however the objectives are weighted by the expected accuracy (in this case, 50 cm based on literature) to avoid overfitting to observation uncertainties. CryoSat-2 has a very long return period but a high spatial resolution therefore we reference the amplitudes to a mean elevation over small river stretches. Because of these simplifications and challenges, the comparison to CryoSat-2 observations is merely a qualitative validation of the water height simulated by the river.

P 3 Lines 25-29: About the previously used models, lumped models have proved less efficient to represent the two annual flood peaks of equatorial rivers, mainly due to a very approximative estimation of PE (Paturel et al 2003) (Dezetter et al 2008) Paturel, J.E., Ouedraogo, M., Mahe, G., Servat, E., Dezetter, A., Ardoin, S. (2003). The influence of distributed input data on the hydrological modelling of monthly river flow regimes in West Africa. Hydrological Sciences Journal, 48, 6, 881-890. Dezetter, A., Girard, S., Paturel, J.E., Mahé, G., Ardoin-Bardin, S., Servat, E. (2008). Simulation of runoff in West Africa: Is there a single data-model combination that produces the best simulation results ? Journal of Hydrology, 354, 203-212.

REPLY: We thank the referee for the comment and references, it is true that the importance of adequately estimating PET is often overlooked.

Plan for revision: We will add a comment on this issue in the introduction along with a reference to the suggested papers, addressing how an adequate estimation of PET may be a limiting factor in some cases.

P 4 Line 29: Hydrological monitoring efforts "by ORSTOM hydrologists during the 50's to the 80's"

REPLY: Thank you for the clarification, the sentence will be updated.

P 4 Line 30: ". . .available informations are from 1984" for most stations, (Mahe et al., 1990; 1994) Mahé, G., Lerique, J., Olivry, J.C. (1990). L'Ogooué au Gabon. Reconstitution des débits manquants et mise en évidence de variations climatiques à l'équateur. Hydrologie Continentale, Ed. ORSTOM, Paris, 5, 2, 105-124. Mahé, G., Delclaux, F., Crespy, A. (1994). Elaboration d'une chaîne de traitement pluviométrique et application au calcul automatique de lames précipitées (bassinversant de l'Ogooué au Gabon). Hydrologie Continentale, 9, 2, 169-180.

REPLY: Thank you for the comment. It is true that this is not the case for all stations, however only about half of the stations have observations dating until the 1980s and only two until 1984 to our knowledge. If the referee has knowledge or access to more recent observations from the basin, we would be very interested in hearing about it and grateful if it can be shared.

Plan for revision: A clarification will be added to the sentence.

P 4 Line 32: there is much more in the paper of Mahe et al. 2013 (update of the 1990's paper), for instance the dramatic reduction of the Spring flood at Lambarene since the 80's, confirmed during the 2000's as showed in the 2013's paper.

REPLY: We thank the referee for the comment and completely agree: although the paper is a region-scale investigation, more details are available concerning the Ogooué

as well.

Plan for revision: We will update the paragraph with additional details from the paper of Mahe et al. (2013), particularly their findings concerning the change in the spring flood since the 1980s and conclusions drawn from the new observations from 2000-2007 at Lambaréné. The paper by Mahe et al. from 1990 will also be referenced in this section of the article.

P 5 Figure 1: the text in white is difficult to read

Plan for revision: The figure text will be made more visible e.g. by changing the color and adding a white text buffer.

P 6 Line 7: historical precipitations at four locations: which ones?

REPLY: We had access to historical precipitation observations at four locations in the basin: Booué (1948-1980), Fougamou (1950-1980), Lebamba (1954-1974) and Petit Okano (1954-1976).

Plan for revision: The location and time of observation at the four locations made available by our project partners will be added to the text.

P 9 Figure 3: too small

Plan for revision: The figure will be enlarged

P 11 Line 1: the storage constants are fixed how? And at which value?

REPLY: The storage constants are spatially and temporally uniform within each calibration zone but are modified during the calibration.

Plan for revision: the sentence will be modified for clarity.

P 11: 3.7 Watershed Delination: Why not used the existing delineation available at the SIEREM website? This site is cited by the authors, but it is difficult to know for what purpose it is cited. http://www.hydrosciences.fr/sierem/index_en.htm

http://www.hydrosciences.fr/sierem/consultation/consultationgraphbas.asp?basid=OGOOUE http://www.hydrosciences.fr/sierem/produits/gis/Ogooue.asp free GIS files soil WHC for 1 2 square degrees, from the FAO soil map of the world. Gives the water height for the upper soil layer. Please cite Boyer et al., 2006 to refer to SIEREM Boyer, J.F., Dieulin, C., Rouché, N., Crès, A., Servat, E., Paturel, J.E., Mahé, G. (2006). SIEREM: an environmental information system for water resources. In: Water Resource Variability: Hydrological Impacts. Proc. of the 5th FRIEND World Conference, La Havana, Cuba, IAHS Publ. 308, 19-25.

REPLY: We thank the author for the comment and reference. The SIEREM website has been used to inform which observations are available in the basin. We make our own watershed delineation in order to ensure that cites of interest and observation stations are resolved by the model. The delineation is very similar to the one provided at the SIEREM website. Regarding the FAO soil map, the maximum soil storage parameter in the model is aggregated vertically and horizontally making a direct comparison tricky.

Plan for revision: The citation of the SIEREM website will be updated and modified to include the Boyer et al. (2006) paper.

P 19 Table 4: the caption mentions number between parenthesis, but there are none in the table. Please clarify.

REPLY: We thank the referee for noticing this inconsistency.

Plan for revision: The values will be added in the table along with a reference to Figure 1 for clarity.

P 19 Line 7: total water storage 70.6 and 83%. OK, but which part of this pcentage participates to surface runoff?

REPLY: The deep aquifer storage releases water directly to the river as return flow as shown in Figure 5. Any decrease in the storage is due to the return flow exceeding the recharge of the aquifers. The deep aquifer holds the lion's share in water storage

change because the storage is aggregated to monthly time steps before comparison to the GRACE observations. Most of the low frequency variations are observed in the deep aquifer because of the smaller storage constants. High frequency variations are averaged out at the monthly time scale and dominate in the other storage compartments.

P 25 Line18-19: there are more than a few decades of observations for the Ogooue river at Lambarene, the time series starts in 1929, and some missing years have been reconstructed. See Mahe et al. 1990

REPLY: The text implies that the most recent records at most stations are more than a decade old. The authors did not have access to any observations ulterior to 1984 at any station, but did have decade-long records available at all locations used for calibration and validation.

Plan for revision: The text will be reformulated to avoid confusion.

P 27 Line 6: OK to thank SIEREM, but the authors should refere to the Boyer et al 2006 paper (see up) Reply: We thank the referee for the clarification and the reference will be added in the text in response to the previous comment.

REPLY: We thank the referee for the clarification and the reference will be added in the text in response to the previous comment

REFERENCES:

Michailovsky, C. I., S. McEnnis, P. a M Berry, R. Smith, and P. Bauer-Gottwein. 2012. "River Monitoring from Satellite Radar Altimetry in the Zambezi River Basin." Hydrology and Earth System Sciences 16 (7): 2181–92. doi:10.5194/hess-16-2181-2012.

---

## Author Comment (AC2) · 16 Oct 2017

Response to the review of Anonymous Referee #2, posted 13/10/2017

The paper investigated the use of multi-mission remote sensing data to force, calibrate and validate a lumped conceptual rainfall-runoff model on an ungauged Ogooué river basin in Africa. The paper is clear and well written. I enjoyed to read this study because is well thought out and organized. The Figures are appropriated even if I would prefer bigger (especially Figure 3, 6 and 7). I recommend the publication of the paper after minor changes below specified.

REPLY: We thank the referee for the feedback and comments on the article. We will review the size of the mentioned figures in the final version.

[Figure]

Some references are not properly assigned to the concept. An example is the paper of Berry et al. (2012) mentioned at P1 Line24 and P2 Line 13 to underline the decline of in-situ gauging networks. I think different papers can replace this citation [1, 2, 3]. Please check also the reference Schumann and Domeneghetti (2016) at P25 Line 26.

REPLY: We thank the referee for pointing this out and for the suggestions. We agree that more appropriate references should be cited at the mentioned places.

Plan for revision: Update citations and make sure there is consistency between reference and statement throughout the paper.

Plots a, b, c of Figure 2 are not mentioned and commented in the text. Please description. Moreover, P6 Line 4 "(Figure 2, c and d)" should be replaced with "(Figure 2, d and e)".

REPLY: We thank the referee for noticing this, indeed a reference to the subfigures is missing and the referee is absolutely right about the cross-reference - it will be updated.

Plan for revision: The subfigures will be referenced P6 Line 3: "The spatial and temporal distribution of rainfall is relatively similar (Figure 2, a and b), however (. . .)"

P19 Line 2: ". . . and simulated the two models in the two basin helves". What does the authors mean with "two models"?

REPLY: The "two models" refer to the two versions forced with the two different remote sensing precipitation products, i.e. the FEWS-RFE forced model and the TRMM forced model.

Plan for revision: The sentence will be clarified to avoid confusion.

Table 4: in the caption parenthesis are mentioned but they are not present in the table. Please correct.

REPLY: We thank the referee for pointing this out.

Plan for revision: The numbers will be added to the table along with a reference to Figure 1.

Table 7: the acronym MD is not specified in the text or in the caption.

Plan for revision: The acronym (MD, Mean Deviation) will be specified in the caption.

References: [1] C. Vörösmarty, A. Askew, W. Grabs, R. G. Barry, C. Birkett, P. Döll, B. Goodison, A. Hall, R. Jenne, L. Kitaev, J. Landwehr, M. Keeler, G. Leavesley, J. Schaake, K. Strzepek, S. S. Sundarvel, K. Takeuchi and F. Webster, "Global water data: A newly endangered species," Eos Trans AGU, vol. 82, no. 5, pp. 54–58, Jan. 2001. [2] N. Sneeuw, C. Lorenz, B. Devaraju, M. J. Tourian, J. Riegger, H. Kunstmann and A. Bárdossy, "Estimating runoff using hydro-geodetic approaches," Surv. Geophys., vol. 35, no. 6, pp. 1333–1359, 2014. [3] D. M. Hannah, S. Demuth, H. A. J. van Lanen, U. Looser, C. Prudhomme, G. Rees, K. Stahl and L. M., Tallaksen, "LargeâAËŸ Rscale river flow archives: importance, current ËǦ status and future needs," Hydrol Process., vol. 25, no. 7, pp. 1191–1200, Mar. 2011.
* * *

---

## Editor Comment (EC1) · B. Schaefli (Editor) · 15 Nov 2017

The paper received two reviews that both conclude that the paper is of very good quality and highly relevant for hydrological research in Africa. Both reviewers provide detailed minor comments and the authors have already answered them. I am currently waiting for a third review. Should this review not arrive within the next few days, I will proceed with the decision.

---

## Author Response (AR1)

The page and line numbers refer to the marked-up version of the manuscript

Review by Anonymous Referee #1, posted 27/09/2017

| Reviewer's comment | This paper is of a great interest to the community of hydrologists in Africa. It is a first attempt to estimate the seasonnal river discharge and its interrannual variability of the Ogooue River in Gabon, from satellite data only. The main interest of this project is to build simulated water heights and discharge time series for virtual gauging stations along the river course, while discharge observed time series end in 1984 for most of the stations, and rainfall data are also difficult to update. The satellite data and the methods used are validated against some in situ data series, and show a good capacity to simulate coherent discharge time series for most of the stations, even if the absolute precision remain of several tenth of cm, which is still difficult to use for real time operational alerts. The paper is well organised and written, and the illustrations are appropriate. I only recommend some minor corrections to clarify some points, enlarge the references list to a few uncited papers related to the core of the study (at least Mahe et al. 1990, see below), and correct some minor errors. |
|---|---|
| Author's response | We thank the referee for the feedback and comments on the article. Particularly, we further thank the referee for the suggested references, and for providing the paper by Mahe et al. (1999). Although we were aware of the paper through citations, we had not been able to retrieve it or the paper by Mahe et al. (1990) until recently. |
| Changes | p.5 l.8: References to the paper by Mahe et al. (1999) and Mahe et al. (1990) have been added. |

| Reviewer's comment | P1 Lines 16-17: the abstract indicates that this study is the best current baseline characterization of hydrological conditions in the Ogooué river. It is partly true, if you consider the previous publication of Mahe et al. 2013 which shows monthly discharges for the Ogooue river over the period 2000- 2007 in regard of previous periods until 1990 (the 90's are missing time series). |
|---|---|
| Author's response | Indeed the previous publication by Mahe et al. (2013) also offers a characterization of hydrological conditions based on the most up-to-date in-situ observations of discharge at the Lambaréné station. However, the current study is the first example of a catchment-scale representation of the Ogooué river regime, including multiple fluxes and storages at daily time step, which can serve as a stepping-stone for simulations of scenarios of change in the basin. |
| Changes | No changes made to the manuscript |

| Reviewer's comment | P 2 Line 27: accuracy between 30 and 70 cm: can the author estimate the discharge value error considering this height margin? |
|---|---|
| Author's response | Estimating river discharge from radar altimetry observations (and thus the propagation of uncertainty into discharge values) is quite tricky, as it requires information on river bathymetry |

| | and the establishment of a rating curve. Michailovsky et al. (2012) converted radar altimetry observations to discharge using in-situ observations from field campaigns and historical records and obtained RMSE values ranging from 4.5 to 7.2% of the mean annual discharge amplitude, corresponding to between 19.9 and 69.4 m^3/s for a water level RMSE between 30 and 70 cm relative to the in-situ levels. While this is possible for missions with repeat ground tracks crossing the river line at specific points with relatively short return periods (e.g. 30 days for Envisat), bathymetry observations throughout the entire river would be required in order to apply this to the CryoSat-2 mission, which is highly impractical.

Furthermore, as historical rating curves and bathymetry observations are not available for the Ogooué, we compare the altimetry water height amplitudes observed by radar altimetry directly to the simulated water height amplitudes. Thus, we do not need to estimate discharge. We use Envisat and Jason-2 observations in the calibration and the accuracy of these missions do impact the estimated discharge values, however the objectives are weighted by the expected accuracy (in this case, 50 cm based on literature) to avoid overfitting to observation uncertainties. CryoSat-2 has a very long return period but a high spatial resolution therefore we reference the amplitudes to a mean elevation over small river stretches. Because of these simplifications and challenges, CryoSat-2 observations are only used for qualitative validation of the water height simulated by the model.

Reference:
Michailovsky, C. I., S. McEnnis, P. a M Berry, R. Smith, and P. Bauer-Gottwein. 2012. "River Monitoring from Satellite Radar Altimetry in the Zambezi River Basin." *Hydrology and Earth System Sciences* 16 (7): 2181–92. doi:10.5194/hess-16-2181-2012. |
|---|---|
| Changes | p.2 l. 30-32: Added reference to uncertainty as demonstrated by Michailovsky et al. (2012) which illustrates the resulting error in discharge estimate. |

| Reviewer's comment | P 3 Lines 25-29: About the previously used models, lumped models have proved less efficient to represent the two annual flood peaks of equatorial rivers, mainly due to a very approximative estimation of PE (Paturel et al 2003) (Dezetter et al 2008)

Paturel, J.E., Ouedraogo, M., Mahe, G., Servat, E., Dezetter, A., Ardoin, S. (2003). The influence of distributed input data on the hydrological modelling of monthly river flow regimes in West Africa. Hydrological Sciences Journal, 48, 6, 881-890.

Dezetter, A., Girard, S., Paturel, J.E., Mahé, G., Ardoin-Bardin, S., Servat, E. (2008). Simulation of runoff in West Africa: Is there a single data-model combination that produces the best simulation results ? Journal of Hydrology, 354, 203-212. |
|---|---|
| Author's response | We thank the referee for the comment and references, it is true that the importance of adequately estimating PET is often overlooked. |

| Changes | p.4 l.1-3: Added a comment on the importance of adequate estimates of PET with a reference to the papers mentioned above. |
|---|---|

| Reviewer's comment | P 4 Line 29: Hydrological monitoring efforts "by ORSTOM hydrologists during the 50's to the 80's" |
|---|---|
| Author's response | Reply: Thank you for the clarification. |
| Changes | P5. l.3 Updated accordingly |

| Reviewer's comment | P 4 Line 30: ". . .available informations are from 1984" for most stations, (Mahe et al., 1990; 1994)

 Mahé, G., Lerique, J., Olivry, J.C. (1990). L'Ogooué au Gabon. Reconstitution des débits manquants et mise en évidence de variations climatiques à l'équateur. Hydrologie Continentale, Ed. ORSTOM, Paris, 5, 2, 105-124.

 Mahé, G., Delclaux, F., Crespy, A. (1994). Elaboration d'une chaîne de traitement pluviométrique et application au calcul automatique de lames précipitées (bassinversant de l'Ogooué au Gabon). Hydrologie Continentale, 9, 2, 169-180. |
|---|---|
| Author's response | Thank you for the comment. It is true that this is not the case for all stations, indeed only about half of the stations have observations dating until the 1980s and only two until 1984. |
| Changes | p.5 l.5 Added clarification |

| Reviewer's comment | P 4 l.32: there is much more in the paper of Mahe et al. 2013 (update of the 1990's paper), for instance the dramatic reduction of the Spring flood at Lambarene since the 80's, confirmed during the 2000's as showed in the 2013's paper. |
|---|---|
| Author's response | We thank the referee for the comment and completely agree: although the paper is a region-scale investigation, more details are available concerning the Ogooué as well. |
| Changes | p.5 l.7-10 Updated paragraph with additional details from the paper by Mahe et al. (2013). |

| Reviewer's comment | P 5 Figure 1: the text in white is difficult to read |
|---|---|

| Author's response | Thank you for the comment. The figure will be made more reader-friendly. |
|---|---|
| Changes | Figure 1: Increased font size and changed color to black and lighter elevation color |

| Reviewer's comment | P 6 Line 7: historical precipitations at four locations: which ones? |
|---|---|
| Author's response | We had access to historical precipitation observations at four locations in the basin: Booué (1948-1980), Fougamou (1950-1980), Lebamba (1954-1974) and Petit Okano (1954-1976). |
| Changes | p.6 l.18-20 the location and time of observation at the four locations made available by our project partners added to the text. |

| Reviewer's comment | P 9 Figure 3: too small |
|---|---|
| Author's response | The figure will be enlarged |
| Changes | Figure 3: Font size and marker size increased |

| Reviewer's comment | P 11 Line 1: the storage constants are fixed how? And at which value? |
|---|---|
| Author's response | The storage constants are spatially and temporally uniform within each calibration zone but are modified during the calibration. |
| Changes | p.11 l.6-7 Modification for clarity |

| Reviewer's comment | P 11: 3.7 Watershed Delination: Why not used the existing delineation available at the SIEREM website? This site is cited by the authors, but it is difficult to know for what purpose it is cited. http://www.hydrosciences.fr/sierem/index_en.htm http://www.hydrosciences.fr/sierem/consultation/consultationgraphbas.asp?basid=OGOOUE http://www.hydrosciences.fr/sierem/produits/gis/Ogooue.asp free GIS files soil WHC for 1 2 square degrees, from the FAO soil map of the world. Gives the water height for the upper soil layer. Please cite Boyer et al., 2006 to refer to SIEREM Boyer, J.F., Dieulin, C., Rouché, N., Crès, A., Servat, E., Paturel, J.E., Mahé, G. (2006). SIEREM: an environmental information system for water resources. In: Water Resource Variability: Hydrological Impacts. Proc. of the 5th FRIEND World Conference, La Havana, Cuba, IAHS Publ. 308, 19-25. |
|---|---|
| Author's | We thank the author for the comment and reference. The SIEREM website has been used to |

| response | inform which observations are available in the basin. We make our own watershed delineation in order to ensure that sites of interest and observation stations are resolved by the model. The delineation is very similar to the one provided at the SIEREM website. Regarding the FAO soil map, the maximum soil storage parameter in the model is aggregated vertically and horizontally making a direct comparison tricky. |
|---|---|
| Changes | p.28 l.8 added citation of suggested reference |

| Reviewer's comment | P 19 Table 4: the caption mentions number between parenthesis, but there are none in the table. Please clarify. |
|---|---|
| Author's response | We thank the referee for noticing this inconsistency. |
| Changes | Table 4: Numbers have been removed and all station names are now indicated in Figure 1 instead. |

| Reviewer's comment | P 19 Line 7: total water storage 70.6 and 83%. OK, but which part of this percentage participates to surface runoff? |
|---|---|
| Author's response | The deep aquifer storage releases water directly to the river as return flow as shown in Figure 5. Any decrease in the storage is due to the return flow exceeding the recharge of the aquifers. The deep aquifer holds the lion's share in water storage change because the storage is aggregated to monthly time steps before comparison to the GRACE observations. Most of the low frequency variations are observed in the deep aquifer because of the smaller storage constant and high frequency variations are averaged out in the other storages. |
| Changes | p.20 l.10-p.21 l.2 Added clarification in text. |

| Reviewer's comment | P 25 Line18-19: there are more than a few decades of observations for the Ogooue river at Lambarene, the time series starts in 1929, and some missing years have been reconstructed. See Mahe et al. 1990 |
|---|---|
| Author's response | The text implies that the most recent records at most stations are more than a decade old. The authors did not have access to any observations ulterior to 1984 at any station, but did have decade-long records available at all locations used for calibration and validation. |
| Changes | p.26 l.19 Text has been reformulated to avoid misunderstandings |

| Reviewer's comment | P 27 Line 6: OK to thank SIEREM, but the authors should refere to the Boyer et al 2006 paper (see up) |
|---|---|
| Author's response | We thank the referee for the clarification |
| Changes | See response above |

Response to the review of Anonymous Referee #2, posted 13/10/2017

| Reviewer's comment | The paper investigated the use of multi-mission remote sensing data to force, calibrate and validate a lumped conceptual rainfall-runoff model on an ungauged Ogooué river basin in Africa. The paper is clear and well written. I enjoyed to read this study because is well thought out and organized. The Figures are appropriated even if I would prefer bigger (especially Figure 3, 6 and 7). I recommend the publication of the paper after minor changes below specified. |
|---|---|
| Author's response | We thank the referee for the feedback on the article. |
| Changes | Figure 3: increased font and marker size
Figure 6: increased size and modified graphics and text setup
Figure 7: Removed redundant subplots and increased font size
Also:
Figure 9: Font increased to improve readability |

| Reviewer's comment | Some references are not properly assigned to the concept. An example is the paper of Berry et al. (2012) mentioned at P1 Line24 and P2 Line 13 to underline the decline of in-situ gauging networks. I think different papers can replace this citation [1, 2, 3]. Please check also the reference Schumann and Domeneghetti (2016) at P25 Line 26.

|---|---|
| Author's response | We thank the referee for pointing this out and for the suggestions. We agree that more appropriate references should be cited at the mentioned places. |
| Changes | Updated following references:
p.1 l.21: Awange et al. 2014 to Tanner and Hughes, 2015 (more representative example of a study using hydrological models to obtain information about river basin hydrology)
p.1 l.23: Berry et al., 2012 to Vörösmarty et al., 2001 and Hannah et al., 2011
p.2 l.2: Added Sneeuw et al., 2014 |

| | p.2 l.14: Replaced Berry et al., 2012 with Hannah et al., 2011
p. 26 l.28: updated reference to Domeneghetti et al. 2014 |
| --- | --- |

| Reviewer's comment | Plots a, b, c of Figure 2 are not mentioned and commented in the text. Please description. |
| --- | --- |
| Author's response | Thank you for pointing this out, indeed a reference to the subfigures is missing. |
| Changes | p.6 l.14: Added reference to subfigures: "The spatial and temporal distribution of rainfall is relatively similar (Figure 2, a and b), however (…)" |

| Reviewer's comment | Moreover, P6 Line 4 "(Figure 2, c and d)" should be replaced with "(Figure 2, d and e)". |
| --- | --- |
| Author's response | Thank you for pointing this out, indeed the reference to the subfigures is incorrect. |
| Changes | p.6 l.16: Figure reference updated |

| Reviewer's comment | P19 Line 2: ". . . and simulated the two models in the two basin helves". What does the authors mean with "two models"? |
| --- | --- |
| Author's response | The "two models" refer to the two versions forced with the two different remote sensing precipitation products, i.e. the FEWS-RFE forced model and the TRMM forced model. |
| Changes | p.20 l.3 Changed to "TRMM and FEWS-RFE forced models |

| Reviewer's comment | Table 4: in the caption parenthesis are mentioned but they are not present in the table. Please correct. |
| --- | --- |
| Author's response | We thank the referee for pointing this out. |
| Changes | Table 4: Locations are indicated by name in Figure 1, therefore the mention of the numbers in parenthesis have been removed from the caption. |

| Reviewer's comment | Table 7: the acronym MD is not specified in the text or in the caption. |
| --- | --- |

| | |
|---|---|
| Author's response | We thank the referee for pointing this out. |
| Changes | Table 7: changed to bias for consistency with Table 8 |

Additional changes:

General:

- Punctuation and grammar changed where needed
- Tables made homogeneous

Table 2:

- Corrected unit of Manning's roughness coefficient
- $X\_GW$ given in fraction rather than % for consistency with model setup as will be released with GW-A toolbox (see code accessibility)

Table 3:

- Numbers in subscript removed from Figure 6 and therefore also from Table 3

Figure 9:

- Font increased to improve readability

Table 5

- Specified objective function given in table

[revised manuscript text omitted]